# 🎛 Interleaved Scene Graphs for Text-and-Image Generation Evaluation

**Dongping Chen**[1,2*], **Ruoxi Chen**[2*], **Shu Pu**[2*], **Zhaoyi Liu**[3*], **Yanru Wu**[2*], **Caixi Chen**[2*],
**Benlin Liu**[1], **Yue Huang**[4], **Yao Wan**[2], **Pan Zhou**[2], **Ranjay Krishna**[1†]

[1]University of Washington, [2]Huazhong University of Science and Technology,
[3]University of Illinois Urbana-Champaign, [4]University of Notre Dame
[*]Equal Contribution, [†]Corresponding Author

**https://interleave-eval.github.io**

## ABSTRACT

Many real-world user queries (*e.g.*, *"How do to make egg fried rice?"*) could benefit from systems capable of generating responses with both textual steps with accompanying images, similar to a cookbook. Models designed to generate interleaved text and images face challenges in ensuring consistency within and across these modalities. To address these challenges, we present ISG, a comprehensive evaluation framework for interleaved text-and-image generation. ISG leverages a scene graph structure to capture relationships between text and image blocks, evaluating responses on four levels of granularity: holistic, structural, block-level, and image-specific. This multi-tiered evaluation allows for a nuanced assessment of consistency, coherence, and accuracy, and provides interpretable question-answer feedback. In conjunction with ISG, we introduce a benchmark, ISG-BENCH, encompassing 1,150 samples across 8 categories and 21 subcategories. This benchmark dataset includes complex language-vision dependencies and golden answers to evaluate models effectively on vision-centric tasks such as style transfer, a challenging area for current models. Using ISG-BENCH, we demonstrate that recent unified vision-language models perform poorly on generating interleaved content. While compositional approaches that combine separate language and image models show a 111% improvement over unified models at the holistic level, their performance remains suboptimal at both block and image levels. To facilitate future work, we develop ISG-AGENT, a baseline agent employing a *"plan-execute-refine"* pipeline to invoke tools, achieving a 122% performance improvement.

## 1 INTRODUCTION

With the proliferation of multimodal language models, it has become apparent that users want models that can simultaneously generate both texts as well as images (Huang et al., 2016; Miech et al., 2019). Consider a scenario where a user asks *"How to make egg fried rice?"* (Figure 1). Answering this query in language - with a list of steps - is one reasonable answer. A more ecological response would follow the style of cookbooks; i.e., by creating intermediate images of the cooking process alongside those steps. Enabling such *multimodal* responses is possible by leveraging a language generation model (Yuan et al., 2022; Gómez-Rodríguez & Williams, 2023) in tandem with a separate image generation model (Rombach et al., 2022; Betker et al., 2023; Blattmann et al., 2023). But the need for dual models slows down inference as both models have to be loaded and run in sequence. Many practical applications, such as writing storybooks (Huang et al., 2016) or generating illustrated instructions (Miech et al., 2019), require generating interleaved images and text.

The community has begun designing unified models with the capability of generating interleaved texts and images for the aforementioned use cases (Zhou et al., 2024a; Li et al., 2024b; Chern et al., 2024). However, generating multiple modalities is challenging. The generations between modalities need to maintain consistency, between multiple images, between multiple sentences, and across the

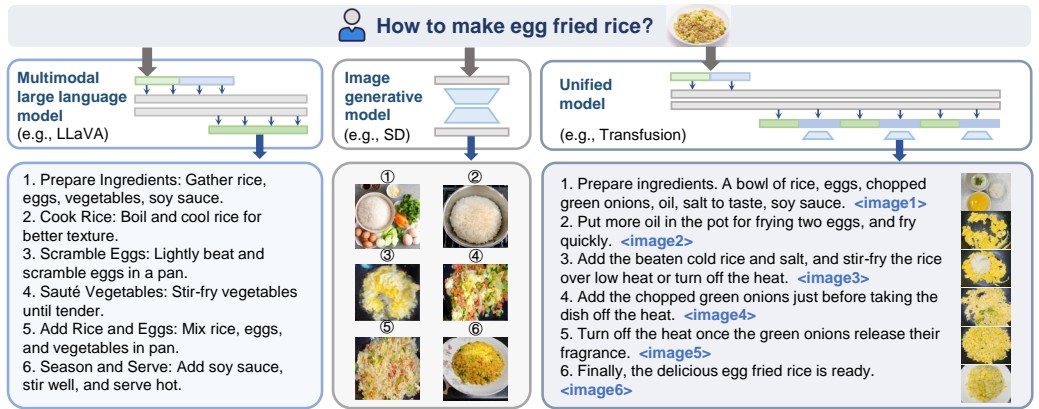

Figure 1: An illustration of differences of each generative model performance on 🖼️**A** (vision-language dominate) tasks, with merely text and image output cannot address the user's problem. See Section 3.2 for how we define 🖼️ (vision dominate) and **A** (language-dominate). **Left:** Text Generation; **Middle:** Image Generation; **Right:** Interleaved Text-and-Image Generation.

generated images and sentences. Benchmarks for such challenges are still in their infancy (Chen et al., 2024e). **1)** Previous benchmarks primarily focus on language-dominate tasks, meaning that queries can be solved with only textual output, thereby not adequately assessing multimodal generation capabilities (Liu et al., 2024d). **2)** The queries in existing benchmarks are free-form without reference answers, making them ambiguous for evaluating multimodal instruction-following generation (An et al., 2023). **3)** Existing benchmarks mainly use an evaluation paradigm called LLM-as-a-Judge (Chen et al., 2024a; Ye et al., 2024), where GPT4 or equivalent model is used for holistic evaluation with their pretrained knowledge (Xia et al., 2024). There is a need for more fine-grained assessment to validate the semantics of each text and image, the consistency between images, the connection between each text and its neighboring image, etc.

We present INTERLEAVED SCENE GRAPH (ISG), an evaluation framework for interleaved image-and-text generation. Conceptually, ISG borrows the scene graph representation as the underlying semantic representation connecting images and text (Krishna et al., 2017; Johnson et al., 2018). ISG automatically parses queries into a scene-graph-like structure, where text and image *blocks* serve as nodes and their relationships as edges. We define a block as a continuous sequence of text or sequence of image tokens. Based on this graph representation, ISG proposes an evaluation protocol across four levels of granularity: holistic (evaluates the entire response in its entirety), structural (evaluates the relationship between blocks), block (evaluates the accuracy within each block), and image (evaluates the contents of an image). The framework translates user queries into (TIFA-like (Hu et al., 2023)) interpretable question answers at each level, enabling systematic and interpretable assessments, and addressing a critical gap in existing research.

Based on ISG, we introduce a benchmark containing user queries with detailed question-answers for evaluating each query across the four levels. ISG-BENCH consists of 8 categories, 21 subcategories classified by their instruction types, and 1, 150 manually collected samples, all incorporating both language-vision dependencies and golden answers to solve the above-mentioned problems. All samples are meticulously collected from previous datasets or built from scratch for high quality. Unlike existing benchmarks, we prioritize *vision-centric* tasks, such as *style transfer*, where the image outputs have specific requirements. Table 1 displays the difference between current interleaved benchmarks and datasets. To validate the accuracy of our evaluation, we compare our automated evaluations with human-annotated judgments across all four levels. ISG shows a Pearson similarity of 0.718 and 0.907, outperforming previous evaluation methods in alignment with humans.

With ISG-BENCH, we evaluate nine accessible interleaved text-and-image generative methods, including five recently popular unified models (*e.g.*, Show-o (Xie et al., 2024), Anole (Chern et al., 2024)), four compositional frameworks (*e.g.*, Claude + SD3 (Esser et al., 2024)). Empirical results demonstrate that current unified models still exhibit significant room for improvement in both instruction following and generation quality. Compositional frameworks significantly outperform unified models in generating high-quality multimodal content, achieving an average holistic score of 6.262 compared to 2.961 from the best unified model (CoMM-MiniGPT-5). However, they still

Table 1: Comparison with existing multimodal interleaved benchmarks. **GT:** Ground truth. **Acc:** Accuracy. **MG:** Multi-granular. 🖼: Image-dominate, **A**: Language-dominate, 🖼**A**: Both.

| Name | #Sample | GT. | Benchmark | | | Evaluation | | | Fine-grained Levels | | | |
|---|---|---|---|---|---|---|---|---|---|---|---|---|
| | | | 🖼 | A | 🖼A | MLLM | Acc | MG | Holistic | Structural | Block | Image |
| MMC4 (Zhu et al., 2024) | - † | ✗ | ✗ | ✔ | ✔ | ✗ | ✗ | ✗ | ✗ | ✗ | ✗ | ✗ |
| CoMM (Chen et al., 2024e) | - ‡ | ✔ | ✗ | ✔ | ✔ | ✔ | ✗ | ✗ | ✔ | ✗ | ✗ | ✔ |
| OpenLeaf (An et al., 2023) | 30 | ✗ | ✗ | ✔ | ✔ | ✔ | ✗ | ✗ | ✔ | ✗ | ✗ | ✗ |
| InterleavedBench (Liu et al., 2024d) | 815 | ✗ | ✗ | ✗ | ✔ | ✔ | ✗ | ✗ | ✔ | ✗ | ✗ | ✗ |
| MMIE (Xia et al., 2024) | 20,103 | ✔ | ✗ | ✔ | ✔ | ✔ | ✗ | ✗ | ✔ | ✗ | ✗ | ✗ |
| GATE OpenING (Zhou et al., 2024b) | 5,400 | ✔ | ✔ | ✔ | ✔ | ✔ | ✗ | ✗ | ✔ | ✗ | ✗ | ✗ |
| **ISG-BENCH (Ours)** | 1,150 | ✔ | ✔ | ✔ | ✔ | ✔ | ✔ | ✔ | ✔ | ✔ | ✔ | ✔ |

†MMC4 contains 101M documents with 571M images.
‡ CoMM contains 227K documents with 2.28M images in both training and test set.

fall short at the block and image levels for accurate generation due to their separate understanding and generation structure, especially in vision-dominated tasks.

Based on the superior performance of compositional frameworks, we propose ISG-AGENT as a compositional baseline for future comparisons. ISG-AGENT generates interleaved text and images through a *"Plan-Execute-Refine"* pipeline (Wang et al., 2024). Specifically, it first produces a plan of tool usage and subsequently executes these advanced tools for interleaved generation, followed by a refinement process for better text-and-image alignment and error fixing. Notably, ISG-AGENT outperforms all other baselines across all four evaluation levels. It achieves an impressive Structural accuracy of 0.871, markedly outperforming the previous best of 0.385 from Gemini. These results underline ISG-AGENT's effectiveness in generating coherent interleaved content, paving the way for more advanced instruction-following agents in multimodal generation and creative applications.

## 2 RELATED WORK

**Interleaved Text-and-Image Generation.** Recent advancements in MLLMs (GeminiTeam, 2023; OpenAI, 2024; 2023; Li et al., 2024a) and diffusion models (Rombach et al., 2022; Esser et al., 2024; Flux, 2024) have led to a surge in research aimed at integrating autoregressive architectures (Liu et al., 2024c; Sun et al., 2024a) for both multimodal understanding (Yue et al., 2024; Li et al., 2023b) and generation tasks (Ghosh et al., 2024; Huang et al., 2023). For understanding, early research has effectively integrated visual perception with pre-trained LLMs using simple visual tokenization (Li et al., 2023a) or projection methods (Li et al., 2023c; 2024a), yielding promising results. Multimodal generation, on the other hand, was initially achieved using pre-trained text-to-image models (Li et al., 2024b; Wu et al., 2023) or through an autoregressive process, where generated tokens are decoded into images (Team, 2024; Chern et al., 2024; Koh et al., 2024). Recently, researchers have started to explore the integration of Transformers and diffusion models, with the aim of unifying multimodal understanding and generation tasks within a single framework (Zhou et al., 2024a; Xie et al., 2024; Wu et al., 2024b), demonstrating potential in interleaved generation of texts and images.

**Automatic Interleaved Text-and-Image Evaluation.** Originating from early text summarization in NLP (Narayan et al., 2018), QA-based evaluation methods automatically transform prompts into questions and use them to validate generated content (Durmus et al., 2020; Deutsch et al., 2020; Eyal et al., 2019). In the multimodal domain, particularly in text-to-image generation, VQA-based evaluation methods transfer text into atomic questions and conduct VQA to verify generated images, providing enhanced fine-grained and interpretable benchmark results (Cho et al., 2023; Lin et al., 2024). Notably, TIFA (Hu et al., 2023) pioneered the use of VQA for automatic evaluation, with multiple subsequent enhancement (Lu et al., 2024; Ghosh et al., 2024; Cho et al., 2024; Chen et al., 2024a). However, evaluating interleaved generations remains challenging. Table 1 shows that existing benchmarks (An et al., 2023; Liu et al., 2024d) heavily rely on zero-shot MLLM-as-a-Judge or traditional metrics (Chen et al., 2024e;b), leading to rough and coarse-grained assessment results.

## 3 INTERLEAVED SCENE GRAPH

We introduce ISG (Figure 2), a comprehensive automatic evaluation framework for interleaved text-and-image generation assessment. Using ISG, we introduce ISG-BENCH, a benchmark for evaluating image-and-text generation.

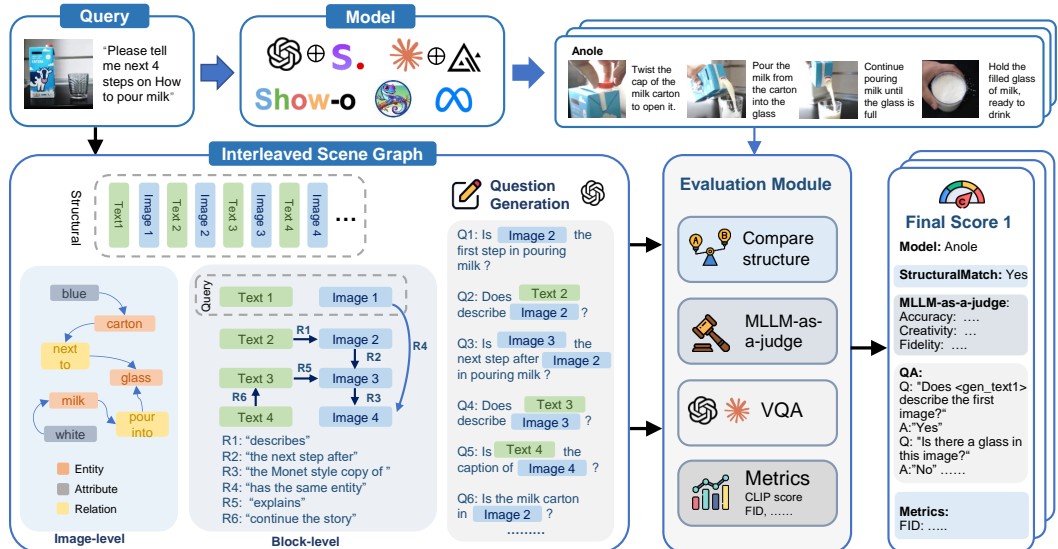

Figure 2: ISG first interprets the user's query into a scene-graph-like structure to enable fine-grained assessment at three levels: **1)** At the structural level, ISG predicts the query's interleaved structure; **2)** At the block level, nodes represent text-image blocks connected by requirement edges; **3)** At the image level, the graph consists of entities, their attributes, and their relationships. Finally, ISG converts each element within the graph structure into questions, evaluates the model's interleaved output using a QA module, and subsequently summarizes these results into a comprehensive assessment.

## 3.1 THE EVALUATION FRAMEWORK

The framework automatically interprets queries into a scene-graph-like structure, where text and image blocks serve as nodes and their relationships as edges. Based on this graph representation, we can perform comprehensively four-level assessment: holistic, structural, block, and image. At each level, the framework generates several question-answer pairs that can be used to evaluate whether a response appropriately answers the query. At the macro level, structural and holistic questions analyze the overall response coherence and quality; while block and image questions assess how accurately each content module adheres to the user's instructions.

- **Structural** questions evaluate whether the response strictly follows the structural requirement in the user's query. As shown in Figure 2, given structural requirement *"generate image first followed by an instruction"*, the correct structure should consist of 4 images interleaved by 4 text blocks. We leverage an LLM to predict the generated structure based on the query and subsequently evaluate answers through direct structural matching.

- **Holistic** questions assess the overall text-image alignment, coherence, and helpfulness by inputting the multimodal query, response, and human-annotated golden answer into an MLLM, which then outputs judgments on the entire answer. Building on previous work (An et al., 2023; Liu et al., 2024d), we enhance the process by employing MLLM-as-a-Judge with golden answers and the *"Analyze-then-Judge"* Chain-of-Thought (CoT) (Wei et al., 2022). This allows for a more human-aligned evaluation, assessing generation quality, text-image alignment, and helpfulness to yield a comprehensive score.

- **Block** questions evaluate fine-grained details within each block. We initially represent the prompt $P$ as subject-object-relation tuples $(sub, obj, r)$, such as $<$ *Text 1, Image 1, Describe* $>$ in the example of Figure 2, where $\{sub, obj\}$ are nodes that denotes image or text block and $r$ is edge that denotes an atomic open-vocabulary requirement. Subsequently, we generate questions from these tuples and evaluate them using the VQA module, with MLLMs providing *"Yes-or-No"* and *"1-10 score"* answers. We also attempt to use CLIPScore (Hessel et al., 2021) for assessing text-image relations, but it fails due to the text block exceeding the text encoder's limit of 77 tokens.

- **Image** questions assess the semantic content of images. We transform multimodal queries into dependency-aware tuples that comprise entities, relations, and attributes, each linked to specific generated images, particularly for vision-dominant tasks such as *"Style Transfer"* and *"Multi-Angle Object"* that have concrete referential answers, whereas the *"Painting"* task requires only

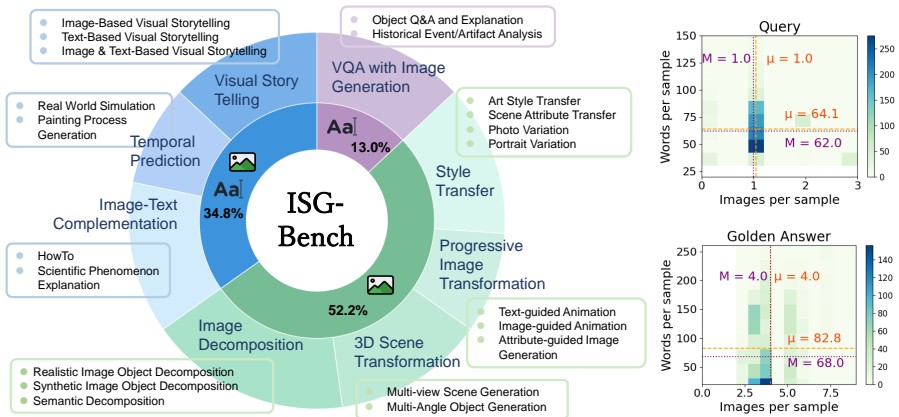

Figure 3: **Left:** An overview of ISG-BENCH. **Right:** Distribution analysis of textual content length and image count for queries and golden answers.

Table 2: Task definitions and additional evaluation dimensions for ISG-BENCH. **Modal:** dominant modality in response evaluation; **Image:** the level of accurate image generation requirements.

| Task | Description | Modal | Image | Subtask | # Sample |
|---|---|---|---|---|---|
| Style Transfer | Generate a sequence of transformed images with corresponding text descriptions. | 🖼 | ⬛ | Art Style Transfer
Scene Attribute Transfer
Photo Variation
Portrait Variation | 50
50
50
50 |
| Image Decomposition | Segment input image into visual elements with text descriptions. | 🖼 | ⬛ | Realistic Image Object Decomposition
Synthetic Image Object Decomposition
Semantic Decomposition | 50
50
50 |
| 3D Scene Transformation | 3D Transformation for images | 🖼 | ⬛ | Multi-view Scene Generation
Multi-Angle Object Generation | 50
50 |
| Progressive Image Transformation | Generate a sequence of images that show gradual changes | 🖼 | ◧ | Text-guided Animation
Image-guided Animation
Attribute-guided Image Generation | 50
50
50 |
| Temporal Prediction | Forecast future or past sequences | 🖼 A | ◧ | Real-world Simulation
Painting Process Generation | 50
50 |
| Image-Text Complementation | Generate complementary visual or textual content | 🖼 A | ▭ | HowTo
Scientific Phenomenon Explanation | 100*
50 |
| Visual Story Telling | Tell a coherent narrative story with images and texts | 🖼 A | ▭ | Image-based Visual Storytelling
Text-based Visual Storytelling
Image & Text-based Visual Storytelling | 50
50
50 |
| VQA with Image Generation | Provide texts and relevant images to answer the question | A | ▭ | Object Q&A and Explanation
Historical Event/Artifact Analysis | 100*
50 |

🖼 denotes accurate image generation requirement for all objects, ◧ for main objects, and ▭ for no requirement.
* For some datasets, we constructed 100 samples because they are more common in life.

the accurate generation of the final image. In contrast, tasks such as *"HowTo"* demand the inclusion of specific objects but allow flexibility in other aspects. We categorize tasks based on the requirement of image generation in the answers as shown in Table 2. These tuples might include *<Image 1, Entity, Cat>* and *<Image 1, Relation, Cat, on the right of, Dog>*. Subsequently, we employ an LLM to generate questions with dependencies and evaluate image generation using these questions via a VQA module (Cho et al., 2023).

For generating VQA questions in block and image levels, we implement ISG with few-shot examples for in-context learning (Dong et al., 2022) and carefully verify these generated questions against human-annotated ground truth. For the evaluation of ISG-BENCH, refer to Section 4.1, and for technical details, see Appendix D.1.

## 3.2 THE BENCHMARK

Based on ISG, we develop the first benchmark, termed ISG-BENCH, for interleaved text-and-image generation to assess multimodal understanding and generation capabilities across various tasks. As shown in Table 2, ISG-BENCH consists of a categorically balanced dataset of 1,150 samples, cov-

ering 21 subtasks across 8 daily interleaved generative scenarios. Each sample includes detailed instructions and structural requirements, such as *"Generate four images and provide a brief text description after your generated image,"* to evaluate both instruction-following capability and interleaved generation ability. Every query is designed to be **1)** vision-language dependent, meaning it cannot be addressed using information from a single modality alone, and **2)** paired with a carefully collected golden reference answer. All samples are collected and manually selected by cross-validation and BERTScore (Zhang et al., 2019) for similarity filtering, as detailed in Appendix B.3.

**Data Collection and Quality Control.** Our benchmark collections involve three main stages. First, we review existing datasets according to the task definition and retrieve high-quality, non-overlapping vision metadata to serve as the visual information in both the query and the golden answer, with some data collected by ourselves (*e.g.*, *"Multi-View Scene Generation"*). We then curate natural language queries that reference the images for automatic evaluation. Each query specifies the required structure of the output. MLLMs are employed to generate textual answers for each task, which are subsequently reviewed by human annotators to ensure accuracy. Due to concerns about data contamination in foundation models (Balloccu et al., 2024; Xu et al., 2024), annotators are instructed to create free-form queries and develop both the query and the corresponding golden answer from scratch. Finally, we obtain a diverse and high-quality interleaved multimodal benchmark with query-answer pairs sourced from various origins. To ensure the quality of our samples, we conduct cross-validation among different annotators for format consistency and typo checking. Detailed definitions, the collection pipeline, and additional examples are provided in Appendix B.

**Modality Specific Assessment.** We categorize each task within our ISG-BENCH into three modes (i.e., Image, Language, and Both) for their primary modality contributing to the output via decision tree (Figure 8). For example, the *"HowTo"* task requires both vision and language content to solve the problem, and *"Art Style Transfer"* requires mainly on vision generation; while *"VQA with Image Generation"* primarily relies on textual output, where the quality and accuracy of answers is mainly attributed to the language component, with generated images serving as complementary information.

## 4 EXPERIMENTS AND ANALYSIS

We first validate ISG against human annotations (Section 4.1), demonstrating its alignment with human judgments. Our subsequent evaluation of interleaved generation (Section 4.2) reveals the limitations of unified models and moderate success of compositional approaches, underscoring current challenges in instruction-following for interleaved generation.

### 4.1 EVALUATING ISG-BENCH

**Experiment Setups.** We leverage one of the most popular MLLMs, GPT-4o (OpenAI, 2024), as question generation and VQA module of our ISG. We conduct experiments to verify the performance of ISG in each step with varying sample sizes and metric settings, as shown in Table 3. Moreover, we verify the *"multimodal-dependency"* of ISG-BENCH in Appendix E.2.

**All results are compared with human-annotated ground truth with cross-validation.** Figure 4 visualizes the distributions of VQA instances in our ISG-BENCH. For the question generation module, we classify a result as correct if it matches the subject and object, with a BertScore (Zhang et al., 2019) higher than 0.8 compared to the ground truth. Our experiments include two settings for VQA module in ISG with an *"Analyze-then-Judge"* COT framework (Wei et al., 2022): *"1-10"* scoring (Lin et al., 2024) and direct *"Yes-or-No"* (Cho et al., 2023). We also conduct ablation experiments on vision inputs or caption images as textual information and few-shot prompting to probe the best setting of ISG. For MLLM-as-a-Judge, we follow previous studies to use human agreement as the evaluation metric (Chen et al., 2024a;f).

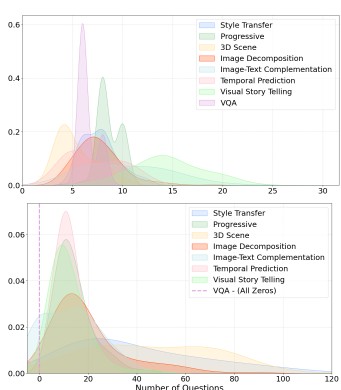

Figure 4: Distributions of VQA instances in Block-level (Upper) and Image-level (Lower).

**ISG demonstrates commendable performance across all tasks in each module.** As illustrated in Table 3, each module of ISG aligns well with human annotation. For structural,

Table 3: When evaluated against human annotations, ISG shows strong alignment with human judgments across all levels. All results for Pearson Similarity have a P-value lower than 0.005. We **bold** better results in two comparative experiments. **Q-Gen:** Question generation module; **Acc+BS:** Accuracy and BertScore for block and question matching respectively.

| Eval Level | Eval Task | Metric | Size | Avg. | 🖼 | | | | 🖼A | | | A |
|---|---|---|---|---|---|---|---|---|---|---|---|---|
| | | | | | Style | Prog. | 3D | Dec. | I-T C. | Temp. | VST | VQA |
| Structural | Direct Match | Accuracy | 1,150 | 1.000 | 1.000 | 1.000 | 1.000 | 1.000 | 1.000 | 1.000 | 1.000 | 1.000 |
| Block | Q-Gen | Acc+BS | 1,150 | 0.967 | 0.955 | 0.988 | 0.890 | 0.970 | 0.993 | 0.980 | 0.980 | 0.980 |
| | VQA Score | Pearson | 1,092 | **0.718** | **0.482** | **0.529** | **0.581** | **0.850** | **0.778** | **0.816** | **0.873** | **0.835** |
| | VQA YesNo | | | 0.446 | 0.169 | 0.386 | 0.528 | 0.382 | 0.555 | 0.388 | 0.634 | 0.529 |
| Image | Q-Gen | Acc+BS | 1,150 | 0.811 | 0.949 | 0.761 | 0.553 | 0.925 | 0.884 | 0.817 | 0.792 | - |
| | VQA YesNo | Accuracy | 4,871 | 0.907 | 0.851 | 0.873 | 0.863 | 0.937 | 0.968 | 0.921 | 0.934 | - |
| Holistic | w. GT | Agreement | 260 | **0.730** | **0.720** | **0.620** | **0.660** | **0.600** | **0.950** | **0.750** | **0.640** | **0.900** |
| | w.o. GT | | | 0.537 | 0.600 | 0.460 | 0.450 | 0.400 | 0.900 | 0.600 | 0.370 | 0.800 |

Table 4: Ablation study on vision input and few-shot help tuple construction in both block-level and image-level. For language-dominate tasks, we do not require accurate image generation.

| Eval Level | Vision | Few-Shot | Avg. | 🖼 | | | | 🖼A | | | A |
|---|---|---|---|---|---|---|---|---|---|---|---|
| | | | | Style | Prog. | 3D | Dec. | I-T C. | Temp. | VST | VQA |
| Block | ✘ | ✘ | 0.631 | 0.635 | 0.801 | 0.495 | 0.778 | 0.725 | 0.621 | 0.787 | 0.207 |
| | ✘ | ✔ | **0.967** | **0.955** | **0.988** | **0.890** | **0.970** | **0.993** | **0.980** | **0.980** | **0.980** |
| | ✔ | ✘ | 0.671 | 0.662 | 0.858 | 0.575 | 0.810 | 0.739 | 0.649 | 0.848 | 0.224 |
| | ✔ | ✔ | 0.942 | 0.934 | 0.959 | 0.822 | 0.969 | 0.981 | 0.970 | 0.949 | 0.954 |
| Image | ✘ | ✘ | 0.688 | 0.873 | 0.751 | 0.497 | 0.908 | 0.575 | 0.526 | 0.684 | - |
| | ✘ | ✔ | 0.804 | 0.902 | **0.796** | 0.518 | 0.905 | 0.869 | **0.859** | 0.780 | - |
| | ✔ | ✘ | 0.711 | 0.943 | 0.755 | 0.535 | 0.951 | 0.586 | 0.539 | 0.671 | - |
| | ✔ | ✔ | **0.811** | **0.949** | 0.761 | **0.553** | 0.925 | **0.884** | 0.817 | **0.792** | - |

ISG exhibits consistent excellence across all tasks, indicating robust potential for capturing structural requirements in interleaved generation instructions. In both Q-Gen and VQA modules, ISG successfully extracts fine-grained requirements with high fidelity to ground truth. For the VQA module, the scoring approach consistently outperforms the *"Yes-or-No"* method, suggesting that more nuanced judgments align better with human evaluations, particularly in ambiguous cases as highlighted in Appendix D.1.1. Vision-guided tasks consistently underperform compared to other tasks, with a noticeable decline in both Q-Gen and VQA modules, underscoring the challenges in automatically evaluating fine-grained aspects of interleaved text-and-image generation. In holistic evaluation, leveraging a golden answer significantly outperforms the zero-shot judging setting of MLLMs, especially in vision-guided tasks, yielding an average of 20% improvement.

**Ablation Study on Vision Input and Few-shot Prompting.** We evaluate our ISG under two conditions: vision input and few-shot examples, for a more comprehensive study. As shown in Table 4, multimodal input varies in block-level and image-level question generation, with a slight enhancement in image-level question generation. In addition, few-shot in-context learning provides dramatic enhancement on both tasks, improving performance by more than 30% in block-level and 10% in image-level tasks, especially in vision-language guided tasks by limiting requirements for the predicted generative content. For language-guided tasks, few-shot learning brings a 70% enhancement in block-level performance, further demonstrating the accurate evaluation framework establishment for this type of creative generation task.

## 4.2 BENCHMARKING INTERLEAVED TEXT-AND-IMAGE GENERATION

**Experiment Setups.** We evaluate 10 frameworks capable of generating interleaved text-and-image content, four recently released unified models, Show-o[1] (Xie et al., 2024), Anole (Chern et al., 2024), Minigpt-5 (Li et al., 2024b), CoMM-Minigpt-5 (Chen et al., 2024e), SEED-LLaMA (Li et al., 2023b) as well as two compositional settings, using Gemini-1.5-Pro (GeminiTeam, 2023) and Claude-3.5-Sonnet (Anthropic, 2024) as a multimodal preceptor[2] and SD3 (Esser et al., 2024) as its generator, with SD2.1 (Rombach et al., 2022) for ablation study. As for ISG, we follow

---

[1]Since Show-o's interleaved generation scripts are unavailable and their current checkpoint lacks multiple-image generation capability, we generate the whole answer via multi-turn dialogues.

[2]Given that AzureOpenAI filters most of our prompt when prompting them to generate caption for image generation, we do not evaluate GPT-4o (OpenAI, 2024) here.

Table 5: Evaluating interleaved text-and-image generation with ISG for structural and holistic level. 🧍 depicts a unified model. 👥 depicts compositional framework.

| | Model | Avg. | 🖼 | | | | 🖼A | | | A |
|---|---|---|---|---|---|---|---|---|---|---|
| | | | Style | Prog. | 3D | Dec. | I-T C. | Temp. | VST | VQA |
| Structural | 🧍 Show-o | 0.295 | 0.320 | 0.253 | 0.380 | 0.000 | 0.195 | 0.700 | 0.080 | 0.433 |
| | 🧍 Anole | 0.000 | 0.000 | 0.000 | 0.000 | 0.010 | 0.000 | 0.000 | 0.000 | 0.000 |
| | 🧍 Minigpt-5 | 0.000 | 0.000 | 0.000 | 0.000 | 0.000 | 0.000 | 0.000 | 0.000 | 0.000 |
| | 🧍 CoMM-Minigpt-5 | 0.000 | 0.000 | 0.000 | 0.000 | 0.000 | 0.000 | 0.000 | 0.000 | 0.000 |
| | 🧍 Seed-Llama-14b | 0.000 | 0.000 | 0.000 | 0.000 | 0.000 | 0.000 | 0.000 | 0.000 | 0.000 |
| | 👥 Claude | 0.323 | 0.000 | 0.000 | 0.030 | 0.760 | 0.313 | 0.500 | 0.000 | **0.980** |
| | 👥 Gemini | 0.385 | 0.005 | 0.093 | 0.000 | **0.959** | 0.453 | 0.549 | 0.107 | 0.913 |
| | 👥 ISG-AGENT | **0.871** | **0.944** | **0.967** | **0.788** | 0.902 | **0.800** | **1.000** | **0.987** | 0.577 |
| Holistic | 🧍 Show-o | 2.329 | 2.112 | 2.407 | 1.434 | 2.868 | 2.056 | 2.578 | 3.315 | 1.863 |
| | 🧍 Anole | 2.810 | 2.931 | 2.764 | 1.850 | 1.485 | 3.209 | 2.575 | 2.968 | 4.695 |
| | 🧍 Minigpt-5 | 2.787 | 2.161 | 3.147 | 1.793 | 2.538 | 2.722 | 2.732 | 2.909 | 4.292 |
| | 🧍 CoMM-Minigpt-5 | 2.961 | 2.602 | 3.085 | 2.237 | 3.090 | 2.523 | 2.720 | 2.874 | 4.557 |
| | 🧍 Seed-Llama-14b | 2.388 | 1.837 | 3.298 | 1.518 | 3.689 | 1.944 | 1.778 | 2.842 | 2.200 |
| | 👥 Claude & SD3 | 6.254 | 5.179 | 6.435 | 3.874 | 7.306 | **7.912** | 5.290 | 6.168 | 7.864 |
| | 👥 Claude & SD2.1 | 5.803 | 4.908 | 4.332 | 3.818 | 6.932 | 7.566 | **5.819** | 5.679 | 7.370 |
| | 👥 Gemini & SD3 | 5.827 | 4.887 | **6.594** | 2.677 | 7.264 | 6.370 | 5.256 | 5.681 | **7.889** |
| | 👥 Gemini & SD2.1 | 5.708 | 5.025 | 6.205 | 2.936 | 7.024 | 6.549 | 4.570 | 5.526 | 7.828 |
| | 👥 ISG-AGENT | **6.262** | **5.873** | 6.459 | **4.887** | **7.582** | 6.932 | 4.540 | **7.030** | 6.795 |
| | 🧍 Human | 9.265 | 9.215 | 9.509 | 9.352 | 8.972 | 9.528 | 9.484 | 9.299 | 8.764 |

Table 6: Evaluating interleaved generation with ISG for block and image level evaluation. We do not report image-level evaluation for language-dominate task *"VQA"*.

| | Model | Avg. | 🖼 | | | | 🖼A | | | A |
|---|---|---|---|---|---|---|---|---|---|---|
| | | | Style | Prog. | 3D | Dec. | I-T C. | Temp. | VST | VQA |
| Block | 🧍 Show-o | 1.962 | 1.719 | 2.087 | 1.351 | 1.000 | 1.632 | 4.421 | 1.233 | 2.252 |
| | 👥 Claude & SD3 | 2.962 | 1.000 | 1.000 | 1.048 | 4.904 | 3.380 | 3.357 | 1.000 | **8.011** |
| | 👥 Claude & SD2.1 | 2.870 | 1.000 | 1.000 | 1.065 | 4.513 | 3.356 | 3.013 | 1.000 | **8.011** |
| | 👥 Gemini & SD3 | 3.081 | 1.018 | 1.500 | 1.000 | **5.077** | 4.204 | 3.533 | 1.434 | 6.885 |
| | 👥 Gemini & SD2.1 | 2.982 | 1.018 | 1.400 | 1.000 | 4.696 | 4.069 | 3.429 | 1.334 | 6.908 |
| | 👥 ISG-AGENT | **5.515** | **5.391** | **6.181** | **6.081** | 4.243 | **6.408** | **6.816** | **5.678** | 3.321 |
| | 🧍 Human | 7.611 | 7.204 | 6.363 | 7.213 | 7.517 | 8.517 | 8.453 | 7.788 | 7.832 |
| Image | 🧍 Show-o | 0.078 | 0.056 | 0.138 | 0.020 | 0.000 | 0.026 | 0.265 | 0.042 | - |
| | 👥 Claude & SD3 | 0.116 | 0.000 | 0.000 | 0.027 | 0.484 | 0.000 | 0.302 | 0.000 | - |
| | 👥 Claude & SD2.1 | 0.104 | 0.000 | 0.000 | 0.014 | 0.432 | 0.000 | 0.281 | 0.000 | - |
| | 👥 Gemini & SD3 | 0.113 | 0.001 | 0.071 | 0.000 | 0.308 | 0.086 | 0.301 | 0.023 | - |
| | 👥 Gemini & SD2.1 | 0.150 | 0.001 | 0.060 | 0.000 | 0.576 | 0.092 | 0.276 | 0.045 | - |
| | 👥 ISG-AGENT | **0.574** | **0.538** | **0.752** | **0.359** | **0.617** | **0.368** | **0.670** | **0.713** | - |
| | 🧍 Human | 0.813 | 0.781 | 0.829 | 0.870 | 0.677 | 0.908 | 0.896 | 0.734 | - |

the best-performed setting in Section 4.1 for a completely automatic evaluating setting. Refer to Appendixes D and E.1 for detailed experiment setups and cost analysis.

**Unified models underperform in accurate interleaved generation.** As illustrated in Table 5, all unified models exhibit significant deficiencies in following our instructions to generate interleaved text-and-image content. Many models produce only one to three images, while some fail to generate any images at all. Consequently, these models could not be subjected to block-level and image-level evaluation protocols. In terms of holistic evaluation, the models demonstrate superior capabilities in language-dominant tasks, while notably underperforming in vision-dominant tasks. This disparity further verifies the hypothesis that current training datasets for unified models lack sufficient vision-dominant instruction tuning samples, such as those for *"Style Transfer"* and *"Image Decomposition"*. Notably, Show-o, as one of the first unified autoregressive models, demonstrates strong structural accuracy but suffers from hallucinations — generating images based on system prompts rather than user instructions, as illustrated in Figure 39. Similarly, Anole achieves SOTA performance among unified models, highlighting the promise of its architectural design.

Table 7: Ablation study for ISG-AGENT in refinement module and advanced tools.

| | Ablation | | Avg. | 🖼 | | | | 🖼A | | | A |
|---|---|---|---|---|---|---|---|---|---|---|---|
| | Refine | SOTA Tools | | Style | Prog. | 3D | Dec. | I-T C. | Temp. | VST | VQA |
| Strcl. | ✗ | ✔ | **0.883** | **0.945** | 0.967 | 0.780 | **0.910** | **0.899** | **1.000** | **0.987** | 0.573 |
| | ✔ | ✔ | 0.876 | 0.944 | 0.967 | **0.788** | 0.902 | 0.840 | **1.000** | **0.987** | **0.577** |
| Block | ✔ | ✗ | 5.155 | 5.013 | 5.615 | **6.833** | 3.229 | 6.031 | 5.784 | 3.541 | **5.198** |
| | ✗ | ✔ | 5.366 | 5.206 | 6.035 | 6.031 | 4.248 | 6.298 | 6.771 | 5.283 | 3.055 |
| | ✔ | ✔ | **5.515** | **5.391** | **6.181** | 6.081 | **4.243** | **6.408** | **6.816** | **5.678** | 3.321 |
| Image | ✔ | ✗ | 0.554 | 0.585 | 0.732 | 0.518 | 0.504 | 0.318 | 0.614 | 0.605 | - |
| | ✗ | ✔ | **0.598** | **0.540** | **0.752** | **0.530** | **0.620** | 0.366 | 0.665 | **0.714** | - |
| | ✔ | ✔ | 0.574 | 0.538 | **0.752** | 0.359 | 0.617 | **0.368** | **0.670** | 0.713 | - |
| Holistic | ✔ | ✗ | 5.433 | 5.477 | 6.024 | 4.544 | 6.630 | 5.971 | 3.980 | 5.585 | 5.256 |
| | ✗ | ✔ | 5.974 | 5.418 | 5.489 | 4.682 | **7.630** | 6.736 | 4.502 | 6.631 | 6.704 |
| | ✔ | ✔ | **6.262** | **5.873** | **6.459** | **4.887** | 7.582 | **6.932** | **4.540** | **7.030** | **6.795** |

**Vision-dominated tasks challenge all models.** Given that these compositional frameworks perceive images and generate images separately, i.e., not end to end, meaning that they naturally cannot perform these tasks well such as accurate image editing due to their inherent structure. On the other hand, although these unified models have the potential to understand and generate images in an end-to-end manner and announce their capability in vision generative tasks such as *"Image Generation"* or *"Image Editing"*, they fall short in understanding multimodal queries to generate interleaved content with multiple images. As shown in Figure 6, the best unified model Anole fails to understand the output format and deviates from the context of input images, demonstrating their deficiency in generating images in vision in-context learning (Sun et al., 2024b).

**MLLM-as-a-Judge cannot evaluate fine-grained accurate generation.** The inconsistency between holistic evaluation results and those at three fine-grained levels, as illustrated in Tables 5 and 6, reveals a notable limitation in MLLM-as-a-Judge to comprehensively assess responses, even when provided with both the user's instruction and correct golden answer. Specifically, MLLM-as-a-Judge struggles to evaluate responses according to fine-grained criteria, such as output structure (including image count) and the detailed text-image relationships stipulated in the prompt. Furthermore, our analysis of the results presented in Table 7 uncovers an inherent bias within MLLM-as-a-Judge, namely *"image-quality bias"*, where higher scores are consistently awarded to responses featuring higher-quality image content, despite these responses potentially violating the user's instructional requirements and judging guidelines. This bias demonstrates that MLLM-as-a-Judge, even provided with a golden answer, still cannot properly perform accurate assessments on interleaved responses that adhere to specified requirements.

## 5 ISG-AGENT: DESIGNING A BASELINE AGENT

Although unified generation models (Chern et al., 2024; Zhou et al., 2024a; Team, 2024) show potential in multimodal interleaved generation, generating interleaved text-and-image content remains challenging, even after fine-tuning. Inspired by previous compositional frameworks for vision generative tasks (Gupta & Kembhavi, 2023; Surís et al., 2023; Ma et al., 2024), we propose ISG-AGENT, a baseline agent for future work to use for the benchmark.

### 5.1 AGENT SETUP

Figure 5 provides an overview of ISG-AGENT, which consists of three components—planning, execution, and refinement—that work collaboratively for interleaved text-and-image generation.

- **Planning.** This component acts as the interface for interpreting the user's multimodal query and generating a corresponding plan for tool usage in JSON format. The plan outlines sequential steps that primarily involve tool invocation. By leveraging an MLLM as the backbone, it ensures the creation of an accurate interleaved generation plan that strictly adheres to the user's instructions, including specifications for fine-grained text-image block requirements. Each step includes clear tool execution functions and natural language descriptions for subsequent tool usage.

- **Tool-usage.** This component is responsible for executing tools with logs (Schick et al., 2024). At each step, it selects the most appropriate tool from the tool library and provides refined, descriptive text and images to the designated tool, such as an MLLM for captioning and diffusion models for

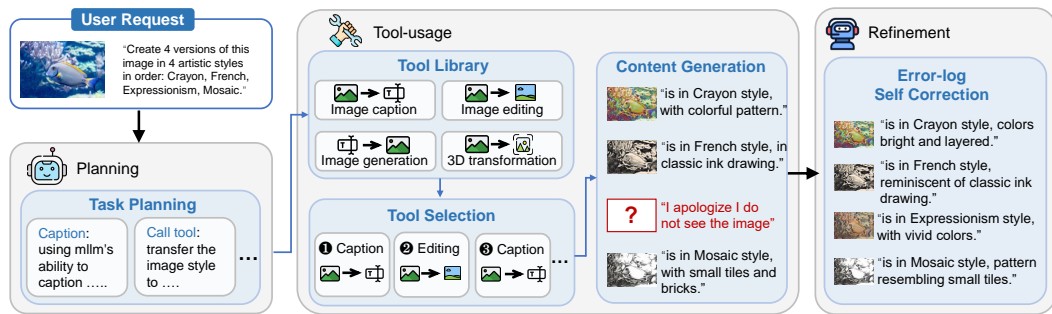

Figure 5: An overview of ISG-AGENT.

image generation. To avoid potential deviations during tool utilization, the agent is designed to generate descriptions that closely align with the instructions specifically for tool-calling.

- **Refinement.** This component is responsible for reviewing and enhancing the quality of the generated content from the previous step by analyzing error messages or improper generation and addressing them by reconstructing the erroneous steps with more detailed and precise execution instructions until the issues are resolved (Wu et al., 2024a). Additionally, this agent refines the text by transforming pronouns, adding conjunctions, and removing repetitive descriptions to improve consistency and textual quality, thus creating more coherent and text-image-aligned content instead of several discrete fragments.

This *"Plan-Execute-Refine"* pipeline for interleaved text-and-image generation ensures that the final output closely adheres to the user's instructions while autonomously handling a variety of tasks effectively. We provide two examples of ISG-AGENT's performance in Figures 37 and 38. For further technical details, please refer to Appendix D.2.

## 5.2 EXPERIMENT

**Setups.** We leverage GPT-4o for planning and verification agent, and use Claude-3.5-Sonnet for tool selector, with SD3 as image generator and multiple tools (UltraEdit (Zhao et al., 2024), DynamiCrafter (Xing et al., 2023), SV3D (Voleti et al., 2024), and DreamMover (Shen et al., 2024)).

**ISG-AGENT outperforms in vision-dominated tasks while falling short in language-guided tasks.** As shown in Table 6, ISG-AGENT strictly follows users' requirements to generate interleaved content, achieving comparative results to human's golden answer in various tasks in both block-level and image-level, especially in vision-dominated tasks like *"Style Transfer"* and *"3D Scene"*. The SOTA results in *"Progressive Transformation"* also demonstrate good coherence of the image content, even accommodate to human-collected answers. Although LLM+Diffusion frameworks fall short in accurate instruction-following, they achieve SOTA results in holistic evaluation in some language-dominate tasks, demonstrating their high generation quality of textual information.

**Enhanced components bring improvement to general response quality.** The comparative analysis between two image generation models (Table 6) and ablation study on tools (Table 7) consistently demonstrates superior performance across various task levels when employing enhanced components, thereby underscoring the importance of advanced tools in producing more accurate and high-fidelity content. Furthermore, the incorporation of a refinement module significantly contributes to improved text-image alignment, substantially enhancing both block-level and holistic performance, which highlights the potential for optimizing individual components to achieve precise interleaved generation within a compositional framework.

## 6 CONCLUSION

This paper advances the field of evaluating interleaved text-and-image generation by introducing the first automatic multi-granular evaluation framework INTERLEAVED SCENE GRAPH and proposing ISG-BENCH with 1,150 multimodal queries over 8 diverse tasks, as well as an agent framework ISG-AGENT for exploring this task. Our comprehensive study, which includes assessments of 10 cutting-edge multimodal interleaved generative frameworks, offers crucial insights and establishes a solid foundation for future research in Appendix A. We emphasize the importance of continued efforts in developing better interleaved generative models and better evaluation frameworks.

**Acknowledgements.** This work is partially funded by Toyota Motor Corporation. We would like to thank Jieyu Zhang, Weikai Huang, and Zixian Ma for their insightful feedback and support.

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

# A DISCUSSIONS AND FUTURE WORKS

**Improving Unified Models with Advanced Interleaved Datasets.** Our results highlight the potential of unified autoregressive model structures like Anole (Chern et al., 2024) and Show-o (Xie et al., 2024), while revealing substantial room for improvement in their instruction following and accurate generation capabilities. This underscores the need for dedicated interleaved datasets, particularly for vision-dominant tasks. Current datasets, limited to unimodal tasks or loosely aligned vision-language pairs (Chen et al., 2024e), inadequately address the challenges of generating coherent interleaved content. Additionally, existing interleaved datasets are predominantly language-centric, failing to establish robust vision-language dependencies crucial for enhanced multimodal understanding and generation. In this context, our compositional agent, ISG-AGENT, shows promise as a pipeline for synthetic interleaved instruction tuning and vision-centric data, potentially advancing the development of unified generative models.

**Improving Evaluation Framework for Transparency and Reliability.** Although we have carefully built the whole benchmark from scratch with cross-validation and evaluated the reliability of these generative models in the question generation and VQA module, concluding that it's practical to use them as evaluators, the potential trustworthiness problem of LLMs should be noted as they still make mistakes in evaluation. Moreover, due to their inherent structure, their evaluation lacks transparent and interpretable results. Therefore, a future direction lies in reducing the AI models in the evaluation process, like *Task Me Anything* (Zhang et al., 2024a), to synthetically generate questions paired with answers to evaluate model performance with highest truthfulness and confidence.

**A Flexible and Integrative Compositional Strategy.** In this study, we explore a compositional agent strategy (Xiao et al., 2024) that integrates diverse model modules to generate interleaved multimodal content. Experimental results indicate that further enhancing each sub-module's performance may significantly improve the overall generative capabilities (Ma et al., 2024). Consequently, the compositional model not only demonstrates high flexibility and adaptability but also serves as a pivotal component in the advancement of unified models, particularly by functioning as a synthetic data pipeline to facilitate interleaved dataset construction. By leveraging high-quality generated content, this synthetic dataset further augments the generalization capabilities of unified multimodal models. Thus, its application not only contributes to exploring the upper-performance bounds of current models but also provides valuable insights and guidance for the design and optimization of future unified models.

**Trustworthiness of Interleaved Generation.** While ISG-BENCH provides a strong foundation for evaluating accurate multimodal interleaved generation, a critical yet underexplored aspect is trustworthiness (Huang et al., 2024) within these models. However, evaluating trustworthiness for interleaved generation presents several key challenges: **1)** Previous research (Liu et al., 2023; Zhang et al., 2023; Huang & Sun, 2023) mainly focus on single-modality generative models (*e.g.*, LLMs), while challenges across text-and-image are not well addressed. **2)** Another significant challenge is assessing the robustness of interleaved generation models against adversarial inputs (*e.g.*, jailbreak attacks (Wei et al., 2024)) or unexpected variations in prompts (Zhu et al., 2023). These models may produce misleading or harmful outputs when manipulated through subtle alterations in the input text or images. Evaluating a model's resistance to such attacks is particularly difficult in a multimodal setting, as an attack could target just one modality (*e.g.*, a slight change in a word or a pixel) and still cause cascading effects on the overall output.

# B DETAILED ISG-BENCH CONSTRUCTION

## B.1 GENERAL INFORMATION

As shown in Figure 3, our benchmark is first categorized by dominated modal, i.e., **Vision**, **Language** and **Both**, followed by 8 categories and 21 sub-categories classified by their definitions. All samples in ISG-BENCH are featuring multimodal input (except one category) with most images collected from existing datasets and text content manually constructed. While MLLMs are used to generate golden answers for some tasks, these underwent thorough human refinement to ensure benchmark accuracy and quality. We provide all MLLM prompts used, and all image-text content was safety-reviewed to ensure benchmark security, quality, and transparency. Refer to Section B.3

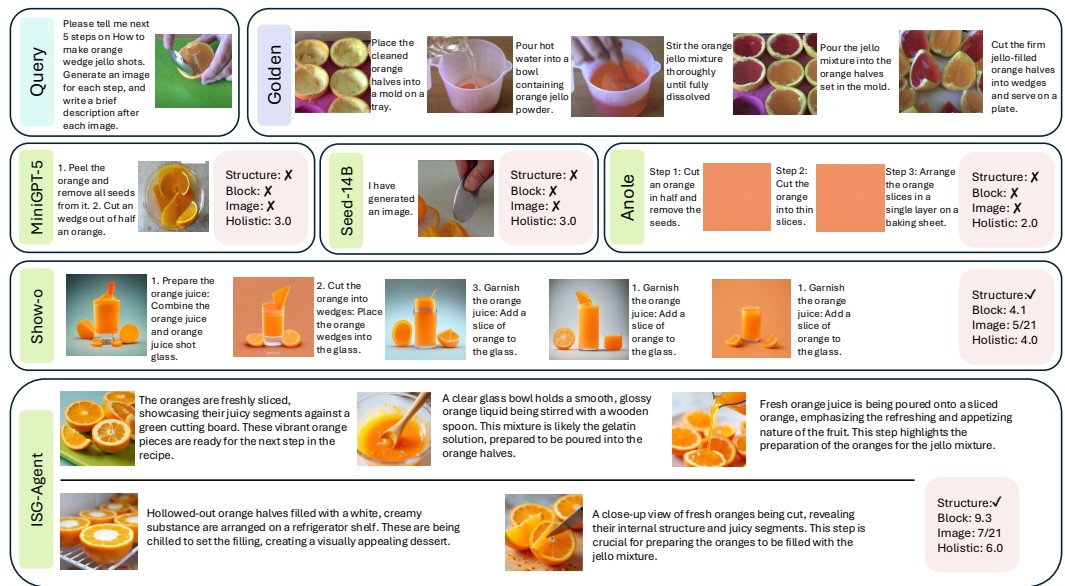

Figure 6: Case study evaluation performed by ISG-BENCH, with each generation resulting to a four-level scoring sheet. Mini-GPT5 and Seed-14B fail to generate interleaved content, while Anole generates low-quality images.

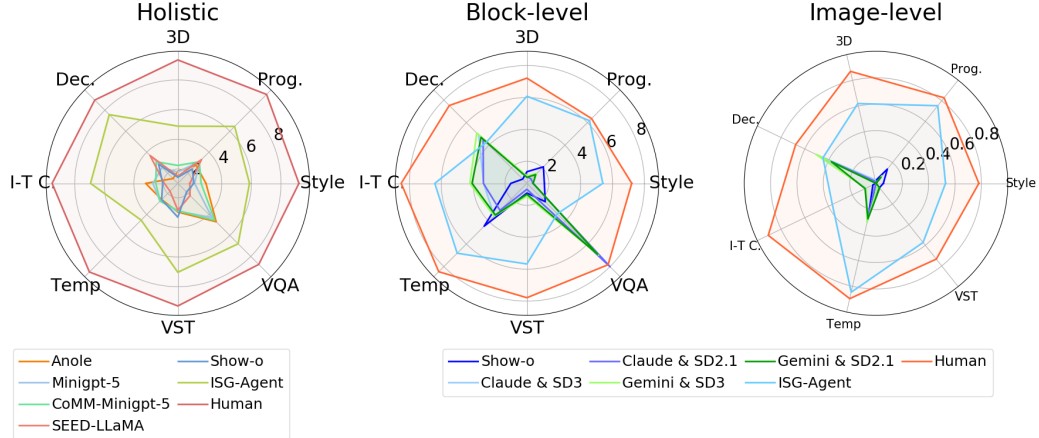

Figure 7: Comparative performance of 🧍 unified models and 👥 compositional frameworks. All interleaved generative methods largely fall behind human-annotated golden answers.

for human annotation details, Section C for additional quantitative analysis, and Section C.2 for NSFW evaluation results.

## B.2 TASK DEFINITION AND SAMPLE COLLECTION

In this section, we provide detailed definitions, collecting pipelines including source datasets and how we collect golden answers, as well as examples for each task in our ISG-BENCH, aiming to provide transparent and detailed construction.

### B.2.1 VISUAL STORY TELLING

This task involves telling a story based on the input. The goal is to generate a coherent narrative sequence that combines both visual and textual information from the image. Previous articles required highly specific design frameworks to accomplish visual storytelling tasks (Liu et al., 2024b; Maha-

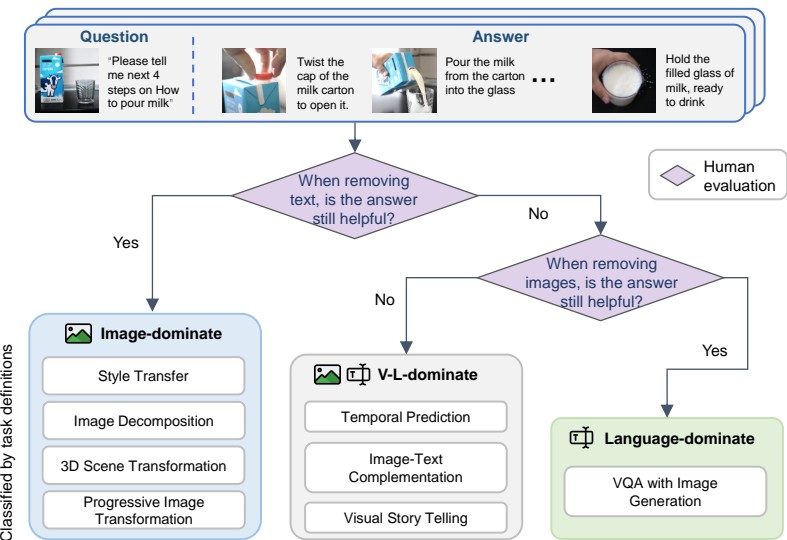

Figure 8: Tasks are classified by task dependency, according to the removal of one modal.

rana et al., 2022; Liu et al., 2024a), whereas the unified model offers a more general framework that is adequate for achieving the same goal.

**Image-based Visual Storytelling.** In this task, we benchmark the capability of image understanding, narrative generation, and creativity by presenting models with input images. Based on its understanding of the input images and its creativity in continuing the story, the model is expected to generate a sequence of image-text pairs. Each image should represent a scene in the story, and each accompanying text should describe the content of the image while also linking it to the preceding and following images. We provide an example of a golden answer in Figure 40.

We utilize images from the StorySalon dataset (Liu et al., 2024a), which offers a rich collection of videos and e-books featuring diverse characters, storylines, and artistic styles. Captions for each image, which include connections to the surrounding context, are generated by GPT-4o using the template shown in Figure 9.

```
Task:  Generate contextually connected captions for each image.
Input:  Images.
Output:  Short captions that describe the storyline depicted in
each image while seamlessly connecting to the surrounding context.
Start with 'image1.', 'image2.'  and so on.
Here are the images:[INSERT_IMAGES]
```

Figure 9: Prompt - Visual Storytelling.

**Text-based Visual Storytelling.** In this task, we benchmark the capability of textual understanding, narrative generation and creativity by presenting models with texts. Based on the input text and its creativity in continuing the story, the model is expected to generate a sequence of image-text pairs. Each image should represent a scene in the story, and each accompanying text should describe the content of the image while also linking it to the preceding and following images. We provide an example of a golden answer in Figure 41.

We utilize images from the StorySalon dataset (Liu et al., 2024a), where each image is accompanied by a short caption. Captions for each image, which include connections to the surrounding context, are generated by GPT-4o using the template shown in Figure 9.

**Image & Text-based Visual Storytelling.** In this task, we benchmark the capability of multimodal understanding, narrative generation and creativity by presenting models with image-text pairs. Based on its understanding of the input images, the text hints for subsequent episodes, and its creativity in continuing the story, the model is expected to generate a sequence of image-text pairs.

Each image should represent a scene in the story, and each accompanying text should describe the content of the image while also connecting it to the preceding and following images. We provide an example of a golden answer in Figure 42.

We utilize images from the StorySalon dataset (Liu et al., 2024a), which offers a rich collection of videos and e-books featuring diverse characters, storylines, and artistic styles. Captions for each image, which include connections to the surrounding context, are generated by GPT-4o using the template shown in Figure 9.

### B.2.2 VQA WITH IMAGE GENERATION

Presented with an image and a question, the model is supposed to not only provide a textual answer but also generate a new, relevant image to support or illustrate its response.

**Object Q&A and Explanation.** In this task, we benchmark the capability of explanatory and knowledge understanding. It involves providing a model with a mixed input of text and an image, where the text includes a question about the image's content. The model is required to identify the subject in the image and generate an interleaved output of text and images that offers a thorough explanation about the subject. Examples can be found in Figure 43.

In this task, we focus on daily object explanations such as animals, plants, insects, daily items and electrical devices. We collect our image data from the Internet using Google and Bing. Provided reference answer to each question is generated by GPT-4o. The prompt to get this reference answer is the same as the original question in our benchmark.

**Historical Event Analysis.** In this task, we benchmark the capability of cultural interpretation, knowledge understanding and visual analysis. This task involves providing the model with a mixed input of text and an image, where the text includes a question about a historical site or artifact depicted in the image. The model is required to identify the place or artifact, describe its historical significance, and generate an interleaved output of text and images that offers a comprehensive analysis. We provide an example of a golden answer in Figure 44.

We collect our image data from the Internet using Google and Bing. Provided reference answer to each question is generated by GPT-4o. The prompt to get this reference answer is the same as the original question in our benchmark.

### B.2.3 TEMPORAL PREDICTION

The model is required to forecast future states or sequences based on initial conditions or partial information, such as predicting the progression of a natural phenomenon or the steps in creating a painting.

**Real World Simulation.** In this task, we benchmark the capability of commonsense reasoning, physical understanding and temporal reasoning. This task involves real-world simulation based on an input image containing both visual elements and text, to generate an image-text sequence that represents physical world phenomena. Each step in the output sequence should include: an generated or modified image showing the progression of the action; and accompanying text describing the change or action taking place. We provide an example of a golden answer in Figure 45.

We use dataset from Panda-70M (Chen et al., 2024d), which contains 70 million high-quality video-caption pairs across various domains, including animals, scenery, and food. We utilize GPT-4o to generate descriptions for images extracted from the relevant videos, with prompts shown in Figure 10.

**Painting Process Generation.** In this task, we benchmark the capability of artistic knowledge and temporal reasoning. This task involves generating a sequence of images and text that simulate the process of creating a painting from start to finish. The model is supposed to produce an image-text sequence that illustrates the painting process, where each step includes: an image showing the current state of the painting; and accompanying text describing the techniques, colors, or elements being added or modified. We provide an example of a golden answer in Figure 46.

We construct a dataset sourced from various painting process videos on YouTube, encompassing a range of painting styles, including oil painting, sketching, quick studies, and digital painting.

```
Task:  Generate a caption for all images except the first one.
Input:  Image sequences about real-world physical phenomena.
Output:  Short captions (10-15 words) describe what is happening
in Image sequences.  Focus on the key changes or actions in
physical world phenomena between images.  Do not include any other
information.
Here are the images:[INSERT_IMAGES]
```

Figure 10: Prompt - Real world simulation.

Additionally, we employ GPT-4o to generate relevant descriptions of each step in the process. The prompt template is shown in Figure 11.

```
Task:  Generate a caption for all images.
Input:  Image sequences about the painting process.
Output:  Short captions (10-15 words) describe the painting stage
in each image.  Focus on the main objects, techniques, or elements
being added or modified between images.  Do not include any other
information.  Start with 'step1:', 'step2:', and so on.
Make sure that the number of your answers is equal to the number of
input images.
Here are the images:[INSERT_IMAGES]
```

Figure 11: Prompt - Painting process generation.

### B.2.4 IMAGE-TEXT COMPLEMENTATION.

The model must generate images based on textual input, or conversely, produce text that complements and explains given images. In this task, visual and textual information are synergistically combined to enhance understanding and communication.

**HowTo.** In this task, we benchmark the capability of sequential reasoning, task decomposition, and procedural understanding. Given a high-level instruction or a text-image pair as input, generate a sequence of image-text pairs that represent steps to accomplish the given task. Each instruction will describe an action or transformation that should occur in the following frames. The output video should be consistent with the provided instructions, maintaining coherent transitions and logical scene progression. We provide an example of a golden answer in Figure 47.

We download HowTo videos from CrossTask (Zhukov et al., 2019) and ChangeIt (Souček et al., 2022), which cover instructional videos collected for different tasks. We captured the frames of the key steps from the video as output images. Descriptions are written by GPT-4o with templates outlined in Figure 12.

```
Task:  Based on the instruction, generate a caption for each image.
Input:  Images sequences and instruction about [INSERT_QUESTIONS].
Output:  Use imperative sentences (10-15 words) describing what is
in each image.  Focus on the main objects and their relationships
between images.  Do not include any other information.  Start with
'1.', '2.'  and so on.
Make sure that the number of your answers is equal to the number of
input images.
Here are the images:[INSERT_IMAGES]
```

Figure 12: Prompt - HowTo.

**Scientific Phenomenon Explanation.**     In this task, we benchmark the capability of scientific knowledge, analytical thinking and procedural understanding. This task requires the model to analyze an image depicting a natural or scientific phenomenon. The model receives a mixed input of text and an image, where the text includes a question about how the phenomenon in the image is formed. The model is expected to identify the phenomenon, describe its formation process, and generate an interleaved output of text and images that offers a detailed explanation. We provide an example of a golden answer in Figure 48.

The data for this task is crafted manually and relevant images are also collected from the Internet using Google and Bing. Reference answers are also generated by GPT-4o with the same prompt as the original question in our benchmark.

### B.2.5   STYLE TRANSFER

This task involves taking an input image with associated text and generating a sequence of transformed images with corresponding text descriptions.

**Art Style Transfer.**   In this task, we benchmark the capability of artistic knowledge, style editing, creativity, and novelty. The model is supposed to generate style-transferred versions of the input image in different art periods (e.g., Renaissance, Impressionism, Cubism), or specified artists (e.g., Van Gogh, Picasso, Monet), each attached with a style description. We provide an example of a golden answer in Figure 49.

We use images from UnlearnCanvas (Zhang et al., 2024c), which includes high-resolution stylized images from 60 different artistic painting styles across 20 different object categories. We directly use the style name of the image to form the description.

**Scene Attribute Transfer.**   In this task, we benchmark the capability of attribute manipulation and image editing. The model is supposed to generate a sequence of image-text pairs, where each image is a transformed version of an input landscape photograph based on specified scene attributes (e.g., weather, lighting, time of day, season), and each text describes the applied transformation. Changes should be photorealistic and faithful to the specified attributes. We provide an example of golden answer in Figure 50.

We use images from TransientAttributes (Laffont et al., 2014), which includes scene appearances with 40 transient attributes related to weather, lighting, time of day, season, and more subjective impressions (e.g. "mysterious" and "soothing"). We manually choose attributes of the image from the dataset to form the description.

**Photo Variation.**   In this task, we benchmark the capability of image analysis and photo editing. The model is supposed to generate a sequence of image-text pairs that show various adjustments to an input photograph, along with descriptive text for each adjustment. Adjustment Categories include exposure, sharpness, brightness, contrast, color temperature, hue, saturation. Changes should be high-quality and natural-looking. We provide an example of a golden answer in Figure 51.

We use photos from MIT-Adobe FiveK (Bychkovsky et al., 2011), which consists of 5 sets of 5,000 example input-output image pairs, each edited by trained photographers. We use GPT-4o to describe images and the prompt template is outlined in Figure 13.

```
Task:  Generate a caption for all images except the first one.
Input:  Images.  Output:  Short captions (5-15 words) describe
what adjustment has been made, when the next followed image is
compared with the first image, one by one.  Do not include any
other information.  Start with '1.  ', '2.'  and so on.
Make sure that the number of your answers is 1 less than the number
of input images.
Here are the images:[INSERT_IMAGES]
```

Figure 13: Prompt - Photo variation.

**Portrait Variation.**   In this task, we benchmark the capability of facial analysis and image editing. The model is supposed to generate a sequence of image-text pairs showing a person at similar

ages based on an input portrait, along with descriptive text for each image. Output images should ensure identity consistency across all generated images. We provide an example of golden answer in Figure 52.

We use human portraits from similar ages of the same person in MORPH dataset (Ricanek & Tesafaye, 2006). Descriptions for each step is written by GPT-4o with following prompt template in Figure 14.

```
Task:  Generate a caption for all images except the first one.
Input:  Images taken in similar ages.
Output:  Short captions (5-15 words) describe what is different,
when the next followed image is compared with the first image, one
by one.  Do not include any other information.  Start with '1.  ',
'2.'  and so on.
Make sure that the number of your answers is 1 less than the number
of input images.
Here are the images:[INSERT_IMAGES]
```

Figure 14: Prompt - Portrait variation.

### B.2.6 IMAGE DECOMPOSITION

This task involves image decomposition based on an input image containing both visual elements and text, with the goal of segment or generating an image-text sequence that breaks down the image into its constituent parts.

**Realistic Image Decomposition.** In this task, we benchmark the capability of object detection and segmentation, and object recognition. The model is supposed to generate image-text pairs where each image showcases objects detected within real-world scenes. The text should detail the objects and the event or relationships between the objects. The output images should ensure the accuracy of the required object present in the image. We provide an example of a golden answer in Figure 53. We selected 50 images from object detection datasets COCO (Lin et al., 2014) and SA1B from Segment-Anything (Kirillov et al., 2023). Rather than using the labeled images directly, we manually identified between two to eight objects in each image to ensure clarity. To maintain task precision, we opted for less crowded scenes. The model is required to generate images that closely resemble the identified objects. Golden answers were crafted using GPT-4o, followed by manual inspection and refinement. The prompt can be found in Figure 15.

```
Task:  Given you the task description and the original image,
generate captions for each object required in the task.  Focus on
objects' key features.
Input:  Task description and the input image.
Output:  You should give feedback in the format required by the
task, first describe the whole image, then orderly caption each
object one by one.  You don't need to generate any image, but
describe them.
```

Figure 15: Prompt - Realistic (synthetic) image decomposition.

**Synthetic Image Decomposition.** In this task, we benchmark the capability of stylized object detection, object identification and extraction. The model is supposed to generate image-text pairs that highlight the detection of objects within virtual or stylized environments, such as digital artwork or fantasy scenes. Each description should caption the corresponding objects in the image. Models should respond without losing any object and precisely cut out the objects from the image or generate similar objects within the image. We provide an example of a golden answer in Figure 54.

We constructed the input images for synthetic image decomposition using search engines and video platforms, resulting in a dataset comprising AI-generated images, real artworks, animations, pixel art, and stamps. To enhance task clarity, we selected images with fewer objects to avoid ambiguity

in descriptions. Similar to the realistic image decomposition task, we utilized the above prompt in Figure 15 to generate reference answers, which were then manually inspected and corrected.

**Semantic Decomposition.** In this task, we benchmark the capability of semantic segmentation and hierarchical understanding. The model is supposed to generate image-text pairs that present the hierarchy of the image. Output Images should precisely segment the region based on the user's prompt from the raw image. The text should correctly label the segmented image and give more information about the image-text input. In addition, enhancing the composition suggestions are given. We provide an example of a golden answer in Figure 55.

We manually selected fifty high-quality and challenging images from the BG-20K dataset (Li et al., 2022) suitable for semantic segmentation. These images encompass not only foreground-background distinctions but also left-right and top-bottom segmentation. To maintain clarity, we avoided overly ambiguous images. We confirmed the segmentation methods with GPT-4o and ultimately constructed golden answers using the following prompt shown in Figure 16.

```
Task:  Given you the task description and the original image,
generate captions for each region required in the task.  Focus on
objects' key features.
Input:  Task description and the input image.
Output:  You should give feedback in the format required by the
task, first describe the whole image, then orderly caption each
region one by one.  You don't need to generate any image, but
describe them.
```

Figure 16: Prompt - Semantic decomposition.

### B.2.7   3D TRANSFORMATION

This task involves 3D Transformation based on an input image containing visual elements and text, with the output being an image-text sequence representing different views or angles of the scene or object.

**Multi-view Scene Generation.** In this task, we benchmark the capability of 3D scene understanding, spatial reasoning and viewpoint synthesis. Based on the input image, the model is expected to generate a sequence of image-text pairs. Each image should depict the scene from the input image as viewed from a different angle, with each accompanying text explaining the observation angle. The objects in the output images must remain consistent across all views. We provide an example of a golden answer in Figure 56.

**Multi-angle Object Generation.** In this task, we benchmark the capability of 3D object reconstruction and single-view to multi-view object synthesis. This task involves providing a model with a single image of an object within a scene, along with a textual instruction that specifies the generation of additional images of the object from different perspectives. The model is required to interpret the instruction and generate a series of images showing the object from various angles, such as left to right, while maintaining consistency in the object's appearance. An instance can be found in Figure 57.

We download images of different angles from ABO (Collins et al., 2022) where we extract golden answer images from five target perspectives. We directly use the angle to form the caption.

### B.2.8   PROGRESSIVE IMAGE TRANSFORMATION

This task involves generating a sequence of images that show gradual changes based on an initial input image and a text prompt. The output is not just a single transformed image, but a series of images showing the progression of the transformation. We gain a 90-images high quality image morphing dataset using Diffmorpher (Zhang et al., 2024b). By dividing them into two parts. One for Text-guided animation, the other for Image-guided animation.

**Text-guided Animation.** In this task, we benchmark the capability of textual understanding and guided image progression modeling. Based on the input image and the accompanying text that

explains the desired final state, the model is expected to generate a sequence of image-text pairs. Each image should represent a stage in the transformation process, and each accompanying text should describe the change occurring in the corresponding part. We provide an example of a golden answer in Figure 58.

We selected 50 easily captioned image pairs from the DiffMorpher dataset (Zhang et al., 2024b). After selection, we prompted GPT-4o to generate captions for a randomly selected image from each pair, replacing the image with a text description as desired final stage. Since the morphing process is an open-ended problem, it aims to make sure each stage is more closer to the text description of the final stage.

**Image-guided Animation.** In this task, we benchmark the capability of visual understanding and state transformation synthesis. Based on the input images, one representing the initial state and the other the final state, the model is expected to generate a sequence of image-text pairs. Each image should depict a stage in the transformation process, with each accompanying text describing the change occurring in the corresponding part. We provide an example of a golden answer in Figure 59.

For this task, we utilized 50 randomly sampled image pairs from the Diffmorpher dataset (Zhang et al., 2024b). To enhance the diversity of the input, we employed data augmentation techniques, which included randomly selecting different initial and final stages. We shifted our focus away from strictly adhering to golden answers. This task aims to make sure each stage is more closer to the final stage's image. Therefore, we pay less attention to the golden answer construction.

**Attribute-guided Image Generation.** In this task, we benchmark the capability of controlled visual attribute manipulation and image synthesis. The model is required to generate a series of images that depict gradual transitions, such as from wealth to poverty, noise to silence, or cleanliness to disorder. Each image reflects the gradual change in state, driven by the transformation or synthesis process while maintaining core structural integrity. The final images should be realistic, clearly illustrating the progression of visual states as guided by the specified changes. We provide an example of a golden answer in Figure 60.

The dataset is created using the DALL-E model from GPT-4o, which generates images based on prompt phrases describing key attribute changes, with accompanying image descriptions also generated by GPT-4o. The prompt template is presented in Figure 17.

```
Task:  Generate a caption for all images except the first one.
Input:   Image sequences about attribute-guided transitions.
Output:   Short captions (5-15 words) describe the key attribute
transitions happening in each image.  Focus on gradual transitions
and changes in the primary attributes.  Do not include any other
information.
Make sure that the number of your answers is 1 less than the number
of input images.
Here are the images:[INSERT_IMAGES]
```

Figure 17: Prompt - Attribute-guided image generation.

### B.3 HUMAN ANNOTATION

The annotation process on ISG is carried out independently by six authors of this paper, each bringing a diverse perspective to the evaluation. Recognizing the importance of annotator diversity, we have selected individuals with varied genders, ages, and educational backgrounds, all of whom possess expertise in the domain. This diversity is instrumental in minimizing bias and enhancing the reliability of our benchmark.

To ensure that our annotators are well-prepared to objectively assess ISG, we have provided them with comprehensive tutorials. These tutorials guide them on how to critically evaluate aspects of the responses, including structure, entities, attributes, and relations. Moreover, we employ cross-validation techniques among different annotators to ensure consistency and objectivity in their judg-

ments. This rigorous approach ensures that our data is marked with a high level of precision and impartiality, providing a robust foundation for our research findings.

We provide an annotation interface for annotation participants, including Image-level VQA human annotation, Block-level VQA human annotation, MLLM-as-a-Judge human agreement annotation and MLLM-as-a-Judge human scoring annotation, as shown in Figure 18.

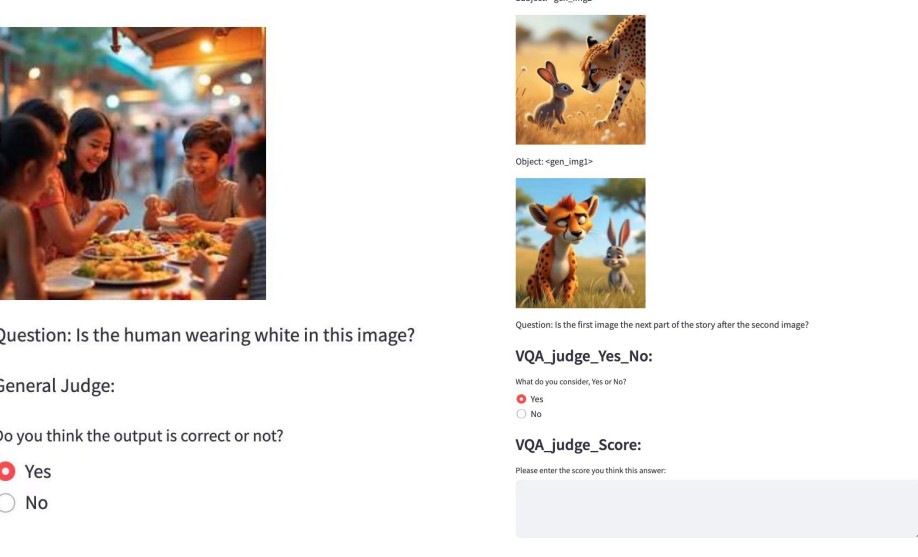

(a) Image-level VQA human annotation
(b) Block-level VQA human annotation

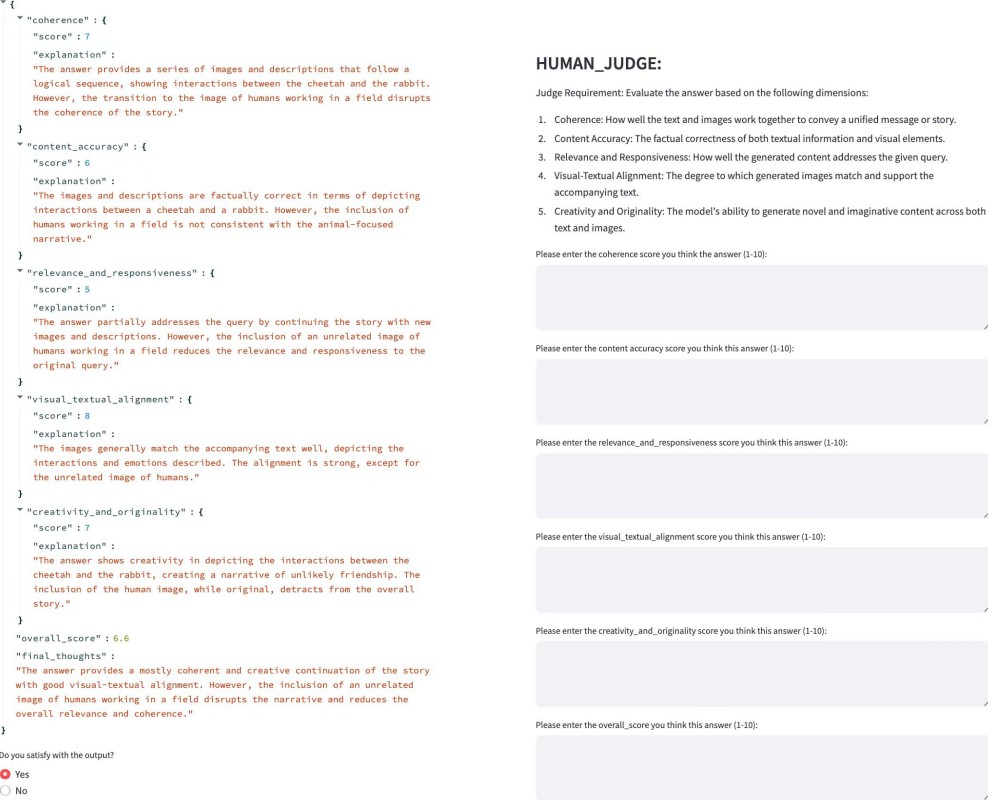

(c) MLLM-as-a-Judge human agreement annotation
(d) MLLM-as-a-Judge human scoring annotation

Figure 18: The annotation interface.

## C   ANALYSIS ON ISG-BENCH

### C.1   GOLDEN ANSWER

Figure 19 illustrates the distribution of image-word number per sample across eight tasks in golden answers. Darker colors indicate a higher number of documents. The $\mu$ and $M$ represent the mean and median number of images/sentences in samples, respectively. These tasks range from *"Visual Storytelling"* to *"Progressive Image Transformation"* with each task having varying distributions of image-to-word ratios.

In tasks like *"Visual Storytelling"* and *"VQA with Image Generation"*, there is a higher concentration of words per sample (above 100), indicating that these tasks require more detailed descriptions. Meanwhile, tasks such as *"Style Transfer"* and *"3D Scene Transformation"*, focus more on images, with a lower median word count. The difference between the mean and median in several tasks also highlights the presence of outliers, particularly in tasks like *"Image Decomposition"* and *"Progressive Image Transformation"*, where a few samples have significantly higher numbers of words or images per sample compared to the majority.

### C.2   SAFETY CHECKING

**This part contains examples of harmful contents. Reader discretion is recommended.**

In this section, we provide a detailed analysis of trustworthiness problems in ISG-BENCH, focusing on NSFW content in images and text content separately. **NSFW Image Filtering.** Figure 20 illustrates the proportion of unsafe and safe images across all categories based on the model's judgments. Out of all the images used, only two (shown in Figure 21) were genuinely unsafe. The remaining images flagged as unsafe were false positives, as demonstrated in Figure 22.

**NSFW Text Content Filtering.** Figure 23 shows the proportions of unsafe and safe text content (both query and golden answer) across all categories, based on the model's evaluations. Among the text content, only one instance in the query and five in the golden answer were truly unsafe. The rest of the flagged text were false positives, as illustrated in Figure 24.

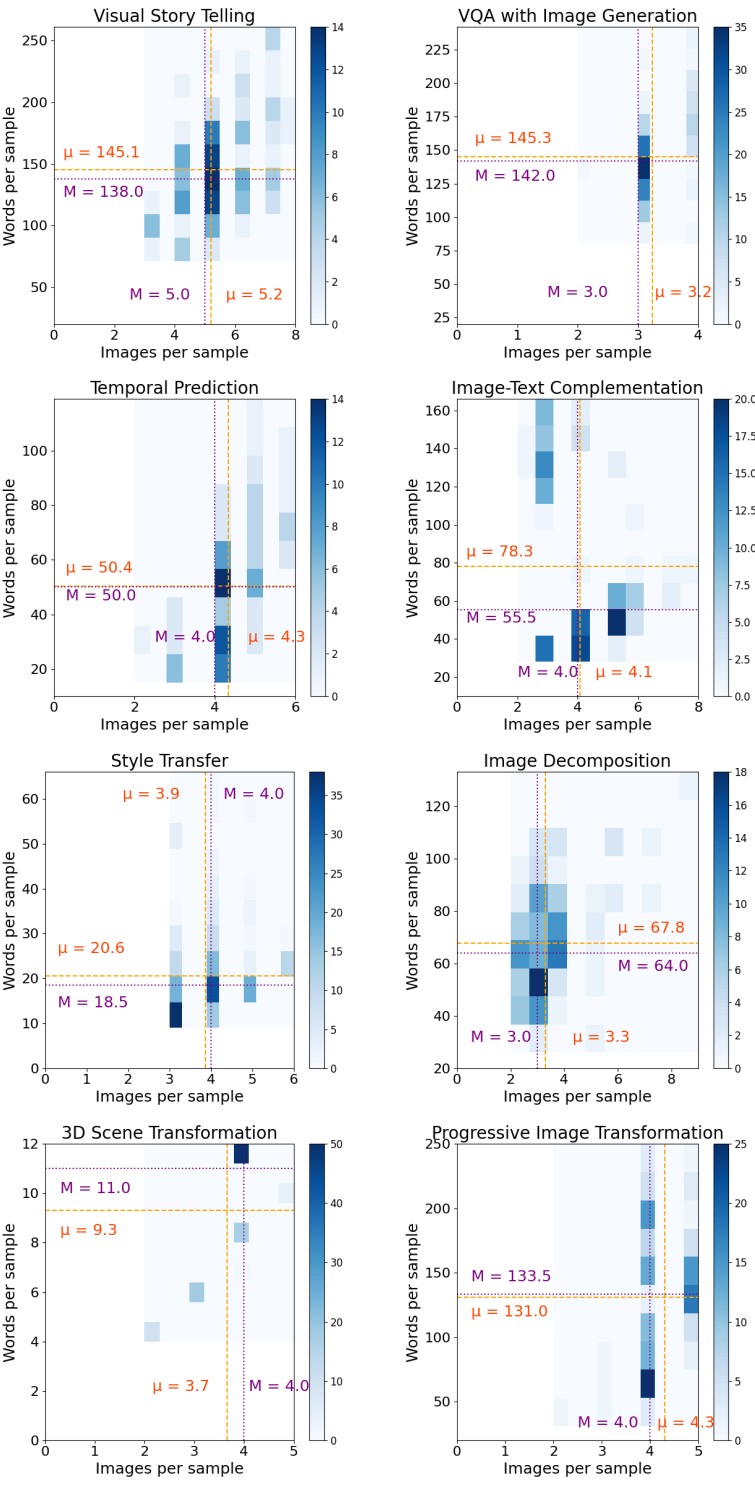

Figure 19: Visualization of the image-word numbers per sample distribution of eight tasks in golden answer. The $\mu$ and M denote the mean/median number of images/words in samples, respectively.

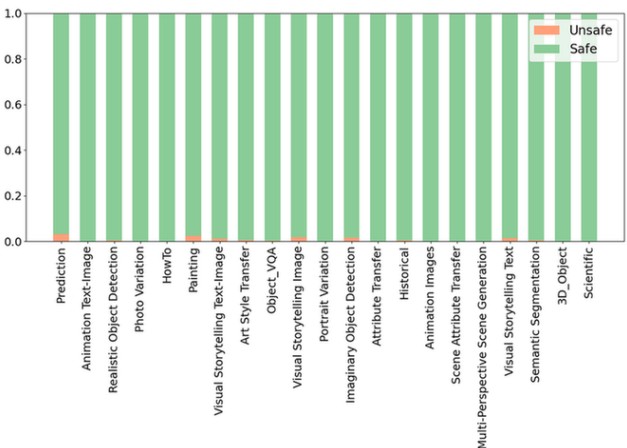

Figure 20: Proportion of unsafe and safe images in each category.

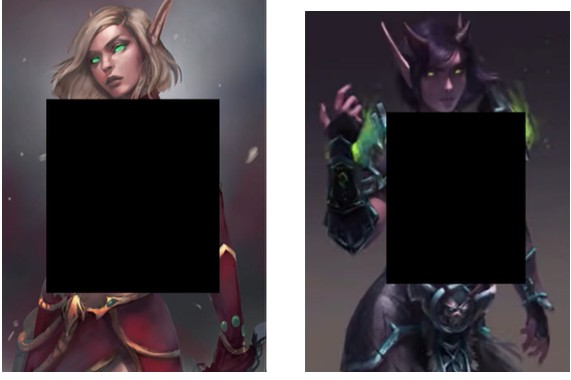

Figure 21: Unsafe images.

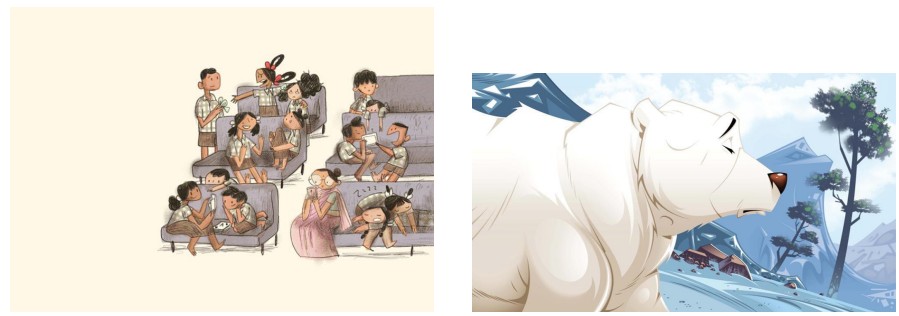

Figure 22: Images that are judged to be unsafe but are actually safe.

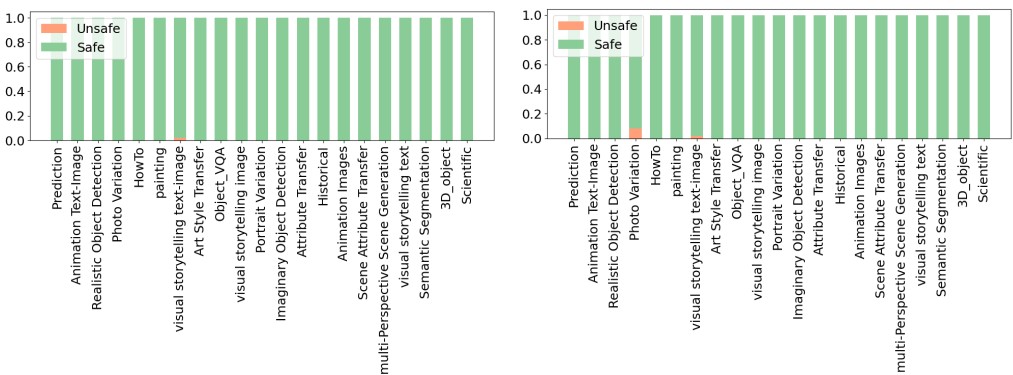

Figure 23: Proportion of unsafe and safe text in each category - query (left) and golden answer (right).

**Query:** "The girl will play happily in the park." Based on the <image1> and the provided text prompts, continue the story, presenting it through 4 new PNG images with descriptions. Ensure each text includes connecting phrases for smooth transitions. For example, <image1> [description1], <image2> [description2].

**Golden Answer:** The little girl cautiously climbs up the tall slide, looking slightly nervous but determined. She reaches the top and sits down, her legs dangling over the edge. She takes a deep breath and pushes off, sliding down the slide with a giggle. She reaches the bottom safely and runs back to the top to do it again.The little boy watches his sister slide down the slide, looking a little jealous. He climbs up the slide and sits down, his legs dangling over the edge. He takes a deep breath and pushes off, sliding down the slide with a giggle. He reaches the bottom safely and runs back to the top to do it again. The little girl and boy continue to slide down the slide, taking turns and having fun. They are safe and happy, enjoying the playground together.

Figure 24: Texts that are judged to be unsafe but are actually safe.

> **Ambiguous Case of ISG**

**Task**: Scientific phenomenon explanation
<**gen_text2**>: As the tsunami travels across the open ocean, it may be barely noticeable on the surface, with wave heights of only a few feet. However, these waves contain enormous energy and can propagate for thousands of miles with minimal loss of energy. The wavelength between crests can be hundreds of kilometers long.

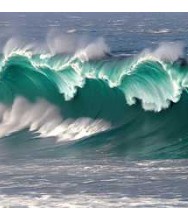

- - - - - - - - - - - - - - - - - - - - - - - - - - - - - - - - - - - - - - - - - -

**Question:** Is this image related to <gen_text2>?
**VQA Yes-or-No**: No
The image shows large ocean waves, but the text describes a tsunami traveling across the open ocean, which is barely noticeable on the surface with wave heights of only a few feet. The image does not depict the described scenario.
**VQA Score**: 8
The image shows large ocean waves, which could be related to the text about tsunamis and their wave characteristics.

Figure 25: Scoring beats Yes/No judging in ambiguous case of ISG.

# D  DETAILED EXPERIMENT SETUPS

In this section, we provide prompts, and the whole pipeline settings of ISG in Section D.1, ISG-AGENT in Section D.2, and models hyperparameter settings in Section D.3.

## D.1  ISG DETAILS

The pseudo-algorithm of ISG is shown in Algorithm 1. We provide prompts for using MLLM to build ISG in and judge model's responses

---

**Algorithm 1** ISG Evaluation

---

1: **procedure** EVALUATE($P$, $G$)                                      ▷ P: Prompt, G: Generated Answer
2:     $S_{\text{Pred}} \leftarrow \text{LLM}(P)$                                      ▷ Predict Structure
3:     **if** ¬StructureMatch($S_{\text{Pred}}$, $G$) **then**
4:         **return** EvaluateWithWhole($P$, $G$)
5:     **end if**
6:     $Questions \leftarrow \text{GenerateQA}(P)$                 ▷ Construct Block-wise (T-T, T-I, I-I) Evaluation
7:     $score \leftarrow 0, total \leftarrow 0$
8:     **for all** $r = (sub, obj, Q) \in Questions$ **do**
9:         $judgement \leftarrow \text{VQA Module}(Q, sub, obj)$
10:         **if** $judgement = $ "Yes" **then**
11:             $score \leftarrow score + 1$
12:         **end if**
13:         $total \leftarrow total + 1$
14:     **end for**
15:     $FinalScore \leftarrow score/total$
16:     **return** $FinalScore$
17: **end procedure**

---

### D.1.1  SCORING EVALUATION REDUCE AMBIGUITY WITHIN VQA MODULE

In VQA, we find that the scoring approach consistently outperforms the *"Yes-or-No"* method, suggesting that more nuanced judgments align better with human evaluations. As shown in Figure 25, the "Yes-or-No" answer is wrong but the VQA score is aligned with human judgement.

**Task:** Extract key information from a multimodal prompt and format it into JSON.
**Input:** A prompt for a multimodal model to generate interleaved text-and-image content. The prompt may include input images.
**Output:** JSON format containing the following keys: 1. "Query": List representing the sequence of images and text in the input 2. "Answer": List representing the sequence of images and text in the expected output
**Special Tokens:** - Input query: - Images: <query_img1>, <query_img2>, ... - Text: <query_text1>, <query_text2>, ... - Generated output: - Images: <gen_img1>, <gen_img2>, ... - Text: <gen_text1>, <gen_text2>, ...
**Here are examples:** ...
**Instructions:** 1. Analyze the given prompt to determine the number of images and text pieces to be generated. 2. Identify the sequence of images and text in both the query and the expected answer. 3. Format the extracted information into the specified JSON structure. 4. There will not be adjacent <gen_text>, such as <gen_textX> <gen_textX+1>. 5. Only output the sequence of images and text noted by <gen_text> and <gen_img> in the "Query" and "Answer". 6. Think before you output your final answer, you can format your thought in a key "Thought" in your output and explain your answer to be generated.
**Here is the prompt:** ...

Figure 26: Prompt - Structure Extraction.

**Task:** Extract and format the relationships between elements in a multimodal prompt and its expected generated answer.
**Input:** 1. Original prompt for a multimodal model 2. Sequence of elements represented by special tokens
**Output:** JSON format with a "relatio" key containing a list of triplets.
**Relation Triplet Format:** (<subject>, <object>, <relation>) - <relation> is an open-vocabulary description (phrase or short sentence) - The triplet should be able to form a fluent English sentence: <subject> <relation> <object> - Avoid duplicate triplets - Only include relations explicitly described in the prompt - The order of <subject> and <object> should reflect the most logical and fluent relationship, regardless of their sequence in the input or output
**Instructions:** 1. Analyze the given prompt carefully. 2. Identify explicit relationships between elements in both the prompt and expected answer. 3. Format relationships as triplets according to the specified format. 4. Ensure the triplets can be easily constructed into fluent English sentences. 5. Use specific descriptors for relations, forming phrases or short sentences. 6. Ensure all triplets are unique and explicitly described in the prompt. 7. Order <subject> and <object> in each triplet to create the most logical and fluent relationship. Do not include relations between input images and texts. 8. Compile the triplets into a list under the "relation" key in the JSON output. 9. Think before you output your final answer, you can format your thought in a key "Thought" in your output and explain your answer to be generated.
**Here are examples:** ...
**Note:** This example includes only the relations that can be confidently inferred from the prompt. The triplets are formed to create fluent English sentences when read as "<subject> <relation> <object>". Do not include any obscure or ambiguous relations that can't be understood by only reading the triplet. For example, "is an image of the third object extracted from" is an obscure relation because there is no "third" within this triplet, you can use "is an image of the object extracted from" instead.
**Here is the prompt:** ...
**Here is the element sequence:** ...

Figure 27: Prompt - Block-Level Requirements Extraction.

### D.1.2 DETAILS OF MLLM-AS-A-JUDGE

We use MLLM-as-a-judge for an overall evaluation, including the following aspects:

- **Coherence**: How well the text and images work together to convey a unified message or story.
- **Content Accuracy**: The factual correctness of both textual information and visual elements.
- **Relevance and Responsiveness**: How well the generated content addresses the given query.
- **Visual-Textual Alignment**: The degree to which generated images match and support the accompanying text.

**Task:** Predict atomic concrete visual entities, attributes, and relations for generated images based on a prompt for a multimodal generative model.
**Input:** 1. Original prompt for a multimodal generative model 2. Sequence of elements represented by special tokens
**Output:** JSON format with a "tuple" key containing a list of tuples in the following formats: 1. Entity: (entity, name of entity, image_id), for example: ["entity", "fish", "<gen_img1>"] 2. Attribute: (attribute, name of attribute, entity, image_id), for example: ["attribute", "yellow", "fish", "<gen_img1>"] 3. Relation: (relation, name of relation, entity1, entity2, image_id), for example: ["relation", "swim in", "fish", "water", "<gen_img1>"]
**Here are examples: ...**
**Instructions:** 1. Carefully analyze the given prompt, focusing on predicting concrete, visual elements that are highly likely to appear in the generated images. Do not describe or analyze any input images mentioned in the prompt. 2. If the prompt includes descriptions or captions of multiple input images, identify common themes and key visual elements across these descriptions. Use these commonalities to inform your predictions for the generated images. 3. Predict tangible entities first. These should be physical objects or beings that can be visually represented in generated images. Avoid abstract concepts or general scenes like 'scene', 'atmosphere', or 'landscape'. 4. For each predicted entity, identify its likely visual attributes. Focus on characteristics that would be visibly apparent in a generated image. 5. You should atomize the entity and attribute as much as possible, and generate entity tuple first, then attribute tuple, and finally relation tuple. Make sure the entities, attributes, and relations are atomic. For example, you should not output "yellow fish" as an entity, you should output "fish" as an entity and "yellow" as an attribute. 6. For attributes and relations, always reference the specific entity or entities they are associated with. 8. Specify which generated image (image_id) each predicted element is expected to appear in. If an entity is likely to appear in multiple generated images, create separate tuples for each image. DO NOT INCLUDE tuples can't be inferred from prompt. 9. Only include tuples for elements you are highly confident (100% Sure) will appear in the generated images based on the prompt and common sense reasoning. Avoid speculating about details that aren't strongly implied by the prompt! 10. For prompts describing a sequence of generated images, consider how visual elements might change or interact across the sequence. 11. Compile the tuples into a list under the "tuple" key in the JSON output.
**Here is the prompt: ...**
**Here is the element sequence: ...**

Figure 28: Prompt - Image-Level Requirements Extraction.

- **Creativity and Originality**: The model's ability to generate novel and imaginative content across both text and images.

## D.2 ISG-AGENT DETAILS

To enable the interleaved generation of multiple image-text pairs, we employ a suite of advanced tools proficient at image generation, editing, and video creation. Our methodology integrates these tools to create a cohesive system capable of producing high-quality visual content alongside descriptive text. Figure 38 and 37 show examples of our ISG-AGENT's output of a randomly selected task from our benchmark.

**Planning Phase.** To improve the model by providing a structured plan, we have designed the following prompt, with a special emphasis on enhancing the model's abilities in:

- **Following Instruction to Output in Format**: Ensuring the model can produce output that strictly adheres to the task required interleaved format.

- **Planning with Tool Box**: Enabling the model to effectively plan tasks by breaking them down into manageable steps, utilizing a set of predefined tools, and providing precise instructions for each step.

- **Enhanced Task Clarity**: Improving the clarity and precision of task instructions, ensuring that each step is well-defined and easy to follow.

- **Consistency in Output**: Ensuring that the output is consistent across different tasks and steps, maintaining a high standard of quality and reliability.

**Task:** Create questions for each provided triplet to verify the stated relationship.
**Input:** A list of triplets in the format (<subject>, <object>, <relation>).
**Output:** A JSON list of objects, each containing the original triplet information and a generated question.
**Here are examples: ...**
**Instructions:** 1. For each input triplet, create an object with the following structure: "subject": "<subject from triplet>", "object": "<object from triplet>", "relation": "relation from triplet", "Question": "<generated question>" 2. Generate a question that, when answered, would verify whether the stated relationship between the subject and object is correct. 3. Ensure the question is clear, concise, and directly related to the triplet's content. 4. Replace image references (e.g., <gen_img1>, <query_img1>) with "this image" if only one image occurs in the triplet, otherwise replace with "first image", "second image" for the subject and object based on their order of appearance in the triplet. 5. Do not use "third" or "fourth" images in the question, as in the question, the maximum number of images could only be 2 (subject and object). 6. Keep text references (e.g., <gen_text1>, <query_text1>) as they are in the original triplet. 7. Frame the question in a way that can be answered with a yes/no or true/false response. 8. Compile all generated objects into a JSON list.
**Notice:** If subject and object are images (<gen_img1>, <query_img1>, etc.), refer the first image as "first image" and the second image as "second image" in your generated question. Ensure that the generated questions are diverse in their phrasing while maintaining clarity and relevance to the original triplet. The questions should be designed to elicit a yes/no or true/false response that verifies the relationship stated in the triplet. Remember to replace image references with "first image", "second image", etc., but keep text references as they are.
**Here is the input:**

Figure 29: Prompt - Block-Level Question Generation.

**Task:** Create questions for each provided triplet of entity, attribute, or relation, and format them into a specific JSON structure.
**Input:** A list of triplets in the following formats: 1. Entity: (entity, name of entity, image_id) 2. Attribute: (attribute, name of attribute, entity, image_id) 3. Relation: (relation, name of relation, entity1, entity2, image_id)
**Output:** A JSON list of objects, each containing the generated question and related information.
**Here are examples: ...**
**Instructions:** 1. For each input triplet, create an object with the following structure: "image": special token refer to generated images, "Question": "<generated question>", "id": <numeric id starting from 0>, "Preliminary": [<list of prerequisite question ids>]
2. Generate a question that verifies the existence of the entity, the presence of the attribute, or the relationship between entities. 3. Ensure the question is clear, concise, and can be answered with a yes/no response. 4. Assign a unique numeric id to each question, starting from 0. 5. Determine any prerequisite questions and list their ids in the "Preliminary" field. - For attributes, include the id of the corresponding entity question. - For relations, include the ids of both entity questions. 6. Compile all generated objects into a JSON list.
**Note:** Ensure that the generated questions are clear and directly related to the triplet's content. The "Preliminary" field should accurately reflect the dependencies between questions, especially for attributes and relations that depend on the existence of entities.
**Here is the input: ...**

Figure 30: Prompt - Image-Level Question Generation.

The prompt is shown in Figure 34 and Figure35, which give concise instruction for planning a task. **Key Instructions** construct the whole planning pipeline.

**Tool Box** offers insights to the planning agent for better step instruction generation. **Considerations** and **Remember** are used for controlling the output into strict format and clear instruction.
The refined plan offers the agent more control over the tool-execution phase.

**Tool-Execution Phase.** We integrate various advanced vision generative models as tools in our ISG-AGENT during agent execution process, detailed as follows:

**Task:** You are given two texts and a question. Please judge whether the question is correct within the two texts.
**Judge Requirement:** You should output a score on a scale of 1-10 and your reason. The score should be a numerical value that reflects how well the question is answered by the given text and image. 10 means the question is answered perfectly. 1 means the question is not answered at all.
**Output Requirement:** Please output in JSON format, directly output your judgment in key "Judge" and your reason in key "Reason". Do not write an introduction or summary. Do not output other irrelevant information.
**Here is the input:**
**Text 1: ...**
**Text 2: ...**
**Question: ...**
Now please judge the question.

Figure 31: Prompt - Block-level VQA - Two Texts.

**Task:** You are a helpful assistant capable of analyzing images and answering questions about them. Your task is to examine the provided image and answer the given question.
**Input:** - An image - A question about the image (e.g., "Is a dog in this image?" or "Is the dog blue in this image?")
**Output:** Provide your response in JSON format with the following structure: "Judge": "Yes" or "No", "Reason": "Your explanation here"
**Instructions:** 1. Analyze the provided image. 2. Consider the question asked about the image. 3. Determine whether the answer to the question is "Yes" or "No". 4. Provide a brief but clear explanation for your judgment. 5. Format your response in the required JSON structure.
**Here is the question: ...**
Now please judge the question. Remember to output in JSON format with "Judge" and

Figure 32: Prompt - Image-level VQA.

- **Image Generation Tool:** We use Stable Diffusion 2.1[3] or Flux.1-dev[4] to generate images based on textual prompts. In the system, the tool agent automatically provides refined and concise prompts extracted from the step's prompt for better generation performance. Input Size: $512 \times 512$ pixels. Inference Steps: 28.

- **Image Editing Tool:** Instructpix2pix[5] and UltraEdit[6] is employed for precise image editing and enhancements. These two models can edit image with step's prompt guidance. We utilize this capability for better instruction-guided image edit. Guidance Scale: 7.5. Image Guidance Scale: 1.5.

- **Video Generation Tool:** DynamiCrafter[7] is a video generation tool under video diffusion framework. It allows both text-image input and images interpolation, which helps to generate temporal continuous and frame-continuous video. In the output, we take screenshots from the video, maintain it's continuity. Input Size: $256 \times 256$ pixels. Video Length: 32 frames. Guidance Scale: 7.5 (unconditional guidance). Guidance Rescale: 0.7. Conditional Input: Text-based input (True).

- **3D Video Generation Tool:** SV3D[8] is utilized for generating 3D video content by only image input. By including this tool, we grant our multi-agents framework the ability to understand the spatial information and to generate views of objects and scenes from different perspectives. In the output, we take fixed frame screenshots to fit our benchmark's tasks. Number of Frames: 21. Conditional Augmentation: 1e-5. Decoding Time: 14.

---

[3] https://stablediffusion.com
[4] https://blackforestlabs.ai/
[5] https://github.com/timothybrooks/instruct-pix2pix
[6] https://ultra-editing.github.io/
[7] https://doubiiu.github.io/projects/DynamiCrafter/
[8] https://sv3d.github.io/

> **Task:** You are a helpful and impartial assistant. You are given a multimodal query with one or several images and a multimodal answer with interleaved text and images. I will also provide a golden answer to the query. Please judge whether the answer is correct and relevant to the query in several dimensions.
> **Judge Requirement:** Evaluate the answer based on the following dimensions: 1. Coherence: How well the text and images work together to convey a unified message or story.
> 2. Content Accuracy: The factual correctness of both textual information and visual elements.
> 3. Relevance and Responsiveness: How well the generated content addresses the given query.
> 4. Visual-Textual Alignment: The degree to which generated images match and support the accompanying text.
> 5. Creativity and Originality: The model's ability to generate novel and imaginative content across both text and images.
> **Output Requirement:** Please output in JSON format, including scores for each dimension (on a scale of 1-10) and a final overall score (on a scale of 1-10). Also provide brief explanations for each score.

Figure 33: Prompt - Overall-level MLLM-as-a-Judge.

- **DreamMover:** DreamMover[9] is an interpolation tool inspired by the DiffMorpher, so we utilize this tool on doing our augemented interpolation or morphing tasks. Guidance Scale: 1.0. Time for Morphing: 3 units.

All the tools are plug-ins within the tool-execution phase. By integrating these tools, we develop a robust pipeline that supports the generation and editing of images interleaved with textual descriptions generated by the ISG-AGENT.

**Refinement Phase.** Refinement Phase's primary capability is to handle errors arising from incorrect planning or tool calling within the Planning-Execution phases. When it fails to generate a final result, it raises an error, which the ISG-AGENT then captures and addresses using different strategies based on the error category. For planning-related errors, the ISG-AGENT detect and reconstructs the entire plan solely based on the original blueprint. In contrast, for other types of errors, it meticulously examines the plan step by step, regenerating relevant input text to provide clearer instructions and ultimately produce an error-free result. Additionally, ISG-AGENT is tasked with smoothing the plan in this phase by creating more continuous text segments, a process facilitated by leveraging the LLM. The prompt used for this operation is illustrated in Figure 36.

**Limitations.** While our agent framework is designed to handle a variety of general interleaved tasks, several limitations must be acknowledged.

- **Lack of Continuity** Despite the Planning Agent's comprehensive understanding of the objectives at each step, the Tool Agent encounters difficulties in accessing information from earlier stages and input accurate information into the tool. Given that large language models operate as black boxes, it is difficult to control the exact prompts input into the Tool Agent, further exacerbating the issue of continuity.
- **Resource Intensity** A significant challenge associated with multi-agent systems is the high resource cost. In our framework, the Tool Agent's effectiveness is heavily dependent on a meticulously planned initial phase. If the planning is not sufficiently detailed or well-structured, the operational efficiency and overall performance of the Tool Agent can be substantially compromised, leading to increased time and computational resource expenditures.

### D.3 MODEL SETTINGS

**Open-source Unified Models.** We employed four open-source unified models, namely Show-o, Mini-GPT5, Anole, CoMM-MiniGPT5 (Mini-GPT5 finetuned on CoMM) and SeedLlama-14B. All hyper-parameters are detailed as follows:

- **Show-o (Xie et al., 2024)** Guidance Scale: 1.75, generation timesteps: 18, temperature: 0.7, resolution: $256 \times 256$.
- **Mini-GPT5 (Zheng et al., 2023).** Image size: 224. temperature: 0.7, repetition penalty: 1.2, guidance scale: 7.5

---

[9] https://github.com/leoShen917/DreamMover

**Task:**
You are a proficient planning agent tasked with writing a step-by-step plan for a 'tool agent' based on a multimodal input. Generate a strictly JSON-formatted plan, ensuring each step leads the tool agent towards a coherent final result.

- - - - - - - - - - - - - - - - - - - - - - - - - - - - - - - - - - - - - - - - - -

**Key Instructions:**

- **Step Format:** Each Step contains a "Task" Category (Only three labels **Call_tool**, **Caption** and **AddImage**), "Input_text" and "Input_images" fields and "Output" field. **AddImage** step only contains "Task" and "Input_images" fields.

- **Control Tool Usage:** All the "Call_tool" steps will be executed at the beginning in order, creating an orderly generated image list [<GEN_0>, <GEN_1>,...], this Task do not affect the final output and the structure because all the generated images will be add to the output in **AddImage** step.

- **Control Result Format:** Design the relationship between **Caption** step and **AddImage** step to fit the structural requirement. **Caption** step will add a text part to the final result, indicating a <gen_text{ID}> placeholder in the structure, while **AddImage** step will add an image fragment to the final result, indicating a <gen_img{ID}> placeholder in the structure. So you should design the order of **Caption** and **AddImage** steps to fit the given plan structural requirement. Do not plan several continuous "Caption" steps, which will be merged into one text fragment in the end.

- **Tool Guidance:** Each **Call_tool** step's text instruction should guide the tool agent on which tool to utilize. Use clear terms like "Generate one image", "Edit the image", "Generate a continuous video", "Generate 3D views", "Morphing from" as needed. Look for the Tool Box for more details.

**Tool Box:**

- **ImageGeneration:** Generates one image based on descriptive text only. No references allowed. ImageGeneration is expert in text-guided image generation, but it cannot see any input image.

- **ImageEdit:** Edits an input image based on a provided prompt and the image. Proficient at editing images like style transfer, attribute modification, and handling subtle changes. When the task requires a change in the input image, use this tool.

- **VideoGeneration:** Creates a sequence of images by input text and one input image guidance. returning several image screenshots of a continuous event. You have to mention how many images you want from this tool. VideoGeneration tool is expert in frame-contiguous and short-time-contiguous generation. Cannot Coexist with other tool in one plan and only can be used once in a task. Do not use this tool to handle subtle changes in image.

- **Video3DGeneration:** Returns multiple chosen views of a 3D object from a single input image. The chosen views should be clearly stated in the Call_tool Input text in the format [Angle1: "Degree-left/right", Angle2: ...]. Only Use this tool when the user wants to retrieve multiple different views of a 3D object or a 3D scene from a single image at once. Cannot Coexist with other tool in one plan and only can be used once in a task. You should provide a list of views you want to get in the input text.

- **ImageMorph:** Return four images of the process of morphing from the first image to the second image. Only Use this tool when the user wants to retrieve the morphing process between two images. Give really simple caption like: 'a photo of [cls]' in the instruction for tool agent to prompt the tool. Cannot Coexist with other tool in one plan and only can be used once in a task. You should provide two images in the input images.

Figure 34: Prompt - Agent planning prompt - Part 1

**Remember:**
Different tool have different input restrictions like any input image, any text input, input image number, ... , and can return different result. You are not doing the task yourself, think of the tool agent! ImageGeneration: input text only; ImageEdit: input text and one image; VideoGeneration: input text and one image; Fixed3DGeneration: input image; Free3DGeneration: input image; ImageMorph: input two images.

**Warning:**

- When your instruction for "Call_tool" is aiming to call 'VideoGeneration', '3DGeneration' or 'ImageMorph', remember these three tools can not coexist with all the other tools in one plan and can only be planned once for the whole task.

- Caption is always executed after all the images are generated, so you should plan any input image in the caption step if necessary.

- Video/3D Generation is less controllable by text so if you want to make text-controllable generation and edit, use ImageGeneration or ImageEdit.

**Considerations:**

- Use explicit terms to avoid ambiguity (e.g., **"Edit the image"** for edits, **"Generate an image"** for new creations).

- Each **Caption** instruction should ask the tool agent to describe every aspect of the image you want to get, instead of generating the caption yourself.

- Each **Call_tool** instruction should be descriptive, focusing on the desired attributes, styles, and settings of the images.

- For sequential tasks, maintain consistency in characters, plot, and style across scenes by instruction.

- When comparing multiple images, ensure the original image is listed first. You should plan a comparison with two or more images in the caption step if the task indicates image comparison or contrast, like multi-perspectives contrast.

- The output should be strictly in **JSON format**, else the extraction will fail.

- **Output placeholders:** For original input images, use #image{ID}#. For all images to be outputted during the process, use <**GEN_ID**> to replace them. For each step's output, use <**WAIT**> as a placeholder. Check the example for more details.

**Note:**
If you want the tool agent to generate an image, your **Input_text** cannot use pronouns like **"previous outline,"** **"original image,"** etc., to refer to any image. Instead, you should provide a detailed description of the image you want to refer to. For example: **"Generate an image of the right cat from the original image. The cat should be white and fluffy, curled up next to a toy."** should be modified to **"Generate an image of a cat. The cat should be white and fluffy, curled up next to a toy."**
**Must:** Your output JSON file must be in officially strict format; any deviation will cause the failure of the evaluation.
**Example Output:**[INSERT_FORMAT]

Figure 35: Prompt - Agent planning prompt - Part 2

---

**Task Instructions for Refining Text Segments in Multimodal Content**

**Sequence and Placeholder Preservation**

- Maintain the exact sequence and number of text segments and image placeholders ('<boi><eoi>') as in the input. No new segments should be added between existing placeholders.

**Text Flow Improvements**

- Rephrase text segments for improved fluency and coherence. Remove redundancy and ensure smooth transitions between segments.

**Image-Text Consistency**

- Integrate references to the images naturally within the text without direct statements about the images. Describe or hint at image content.

**Error Handling**

- If any references to missing or problematic images occur, remove apologies or explanations to keep the narrative smooth.

**Key Instructions:** Output must match input in sequence and number of placeholders. Ensure coherent and engaging text. Eliminate redundancy and fix fragmented sentences.
**Fewshot Example:** [INSERT_FORMAT]

---

Figure 36: Prompt - Agent refinement and verification prompt - Part 1

- **CoMM-MiniGPT5 (Chen et al., 2024e).** Image size: 224. temperature: 0.7, repetition penalty: 1.2, guidance scale: 7.5
- **Anole (Chern et al., 2024).** Text repetition penalty: 0.7 Text temperature: 0.5, Text top_p: 0.8, Text greedy: False; Image guidance scale text: 2.0, Image guidance scale image: 1.2, Image temp: 0.5, Image top_p: 0.7, Image greedy: False.
- **Seed-Llama 14B (Li et al., 2023b).** Temperature: 0.7, num_beams: 1, top_p: 0.5, Image size: 224.

**Other Models.** We utilize three proprietary models, GPT-4o, Claude-3.5-Sonnet, and Gemini-1.5-pro-latest as multimodal preceptors and Flux-dev, SD3, SD2.1 as image generators, with detailed settings as follows:

- **Gemini-1.5-pro-latest (GeminiTeam, 2023).** Temperature: 1, top_p: 0.95
- **Claude-3.5-Sonnet (Anthropic, 2024).** Temperature: 0.9
- **GPT-4o OpenAI (2024).** Temperature: 1, top_p: 1
- **Flux1-dev (Flux, 2024).** Guidance scale: 3.5, num_inference_steps: 50
- **Stable Diffusion 3 (Esser et al., 2024).** Guidance scale: 7.0, num_inference_steps: 28
- **Stable Diffusion 2.1 (Rombach et al., 2022).** Guidance scale: 7.5, num_inference_steps: 25

Example of ISG-AGENT performance on Historical Event Analysis task

**Input**: What is the historical significance of the site or monument depicted in the picture <**image1**>? Provide detailed information about its history, including when its property, its cultural importance, and any relevant events associated with it. You should answer my question in text first and then generate four interleaved sections of text followed by images, where each image matches the text that comes before it. The last block must be image and every image are separated with text content. For example, <**gen_text1**>, <**gen_img1**>, <**gen_text2**>, <**gen_img2**>....

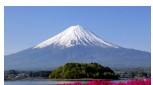

**Agent Response:**

Mount Fuji, an iconic symbol of Japan, stands majestically at 3,776 meters (12,388 feet) tall. Located on Honshu Island, this active stratovolcano has been revered in Japanese culture for centuries. Its near-perfect conical shape has inspired artists and poets, and it remains a popular destination for pilgrims and tourists alike. The mountain's beauty is enhanced by the surrounding landscape, with lakes, forests, and colorful flowers at its base, creating a picturesque scene that epitomizes the natural splendor of Japan.

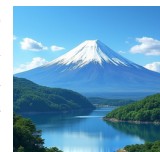

Mount Fuji holds immense cultural significance in Japan. As one of the country's 'Three Holy Mountains', it has long been revered in Shinto religion. The mountain's iconic symmetrical cone shape has inspired countless artists and writers throughout history, becoming a symbol of Japan itself. Its image appears frequently in traditional ukiyo-e woodblock prints, most famously in Hokusai's 'Thirty-six Views of Mount Fuji' series. In literature, Mount Fuji features prominently in classical poetry and modern works alike, often symbolizing beauty, purity, and the spirit of Japan. Its cultural importance led to its designation as a UNESCO World Heritage Site in 2013, recognizing it as a place of artistic and spiritual inspiration.

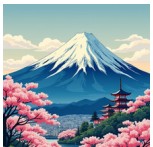

It shows Mount Fuji in its iconic form, surrounded by cherry blossoms and a traditional pagoda. The Hoei Eruption of 1707-1708 was a significant event that temporarily altered Fuji's appearance and impacted surrounding areas. This eruption produced substantial ashfall, affecting agricultural productivity and daily life in nearby regions, including Edo (modern-day Tokyo). The eruption's effects were felt far and wide, with ash deposits reported as far as 100 km away from the volcano.

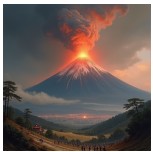

Mount Fuji's iconic silhouette and cultural significance make it a natural choice for UNESCO World Heritage status. Its perfect conical shape and fiery eruption in this image showcase why it's such a powerful symbol of Japan. As a popular climbing destination, thousands ascend its slopes each year to witness the sunrise from its summit. The surrounding area, with its scenic beauty and traditional buildings, attracts tourists year-round who come to admire the mountain from various vantage points and experience Japanese culture in its shadow.

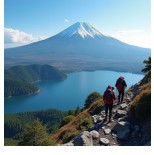

Figure 37: Example of ISG-AGENT performing the Historical Event task

---

**Example of ISG-AGENT performance on How-to Analysis task**

**Input**: I will give you an this image. Based on it, please tell me next next 5 steps on How to make celery juice. Generate an image for each step, and write a brief description after each image. For example, <**gen_img1**>[description1], <**gen_img2**> [description2], <**gen_img3**> [description3], <**gen_img4**> [description4].

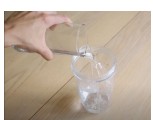

- - - - - - - - - - - - - - - - - - - - - - - - - - - - - - - - - - - - - - - - - - -

**Agent Response:**

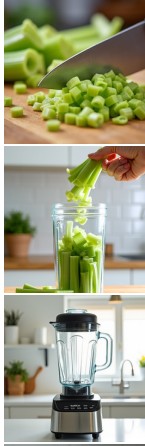

In this step, the celery is being finely diced on a wooden cutting board using a sharp knife. The precise knife skills are evident as the celery is cut into small, uniform cubes. The vibrant green color of the celery contrasts beautifully with the warm tones of the wooden surface, highlighting the freshness of the ingredients.

Here, chopped celery is being added to a tall glass jar. A hand is seen dropping pieces of fresh, bright green celery stalks into the jar, which already contains some chopped celery. The clean kitchen background emphasizes the preparation of a healthy snack or recipe.

A modern blender is ready on the kitchen countertop, featuring a large, clear glass jar with a black lid securely in place. The stainless steel base with a black control panel adds a sleek touch to the kitchen setting, which includes a sink, potted plants, and natural light, creating an inviting atmosphere for blending.

The blender jar is now filled with a vibrant green liquid as the celery stalks are being blended. The swirling, frothy mixture is captured mid-action, with the green juice splashing against the sides of the container. The clear glass provides a perfect view of the blending process, set against a kitchen counter backdrop with additional ingredients.

Finally, the vibrant green liquid is being poured from a container into a clear glass. The smooth, slightly frothy texture of the juice is captured mid-stream, creating a dynamic and appetizing visual. The bright green color stands out against the plain, light-colored background, emphasizing the freshness and healthiness of the juice.

Figure 38: Example of ISG-AGENT performing the How-to task

Table 8: Average token consumption for evaluating 1 sample with ISG-AGENT.

| Level | Output | Input | Total |
|---|---|---|---|
| Image | 280 | 2,285 | 2,565 |
| Block | 738 | 3,221 | 3,959 |
| Holistic | 264 | 1,216 | 1,480 |
| Overall | 1,282 | 6,722 | 8,004 |

Table 9: Average computing time for one sample in ISG-BENCH on A800 servers.

| Category | Anole | Show-o |
|---|---|---|
| Style Transfer | 43.330 | 120.103 |
| Progressive | 47.093 | 91.917 |
| 3D Scene | 33.205 | 73.095 |
| Image Decomposition | 21.830 | 91.990 |
| Image-Text Complementation | 52.280 | 74.335 |
| Temporal Prediction | 33.190 | 53.150 |
| Visual Story Telling | 28.150 | 111.400 |
| VQA | 36.330 | 185.380 |

Table 10: Average tokens and estimate time (600 tokens per second via API) of ISG-Agent in ISG-Bench.

| Category | Avg Input | Avg Output | Time(s) |
|---|---|---|---|
| Style Transfer | 14448 | 1801 | 27.0 |
| Progressive | 14649 | 2019 | 27.8 |
| 3D Scene | 7755 | 1158 | 14.8 |
| Image Decomposition | 12370 | 1802 | 23.6 |
| Image-Text Complementation | 12196 | 1758 | 23.3 |
| Temporal Prediction | 9800 | 1358 | 18.6 |
| VST | 13712 | 2230 | 26.5 |
| VQA | 9128 | 1139 | 17.1 |

# E  ADDITIONAL EXPERIMENT RESULTS

## E.1  COST ANALYSIS

We conducted a cost analysis on a sample test, with results presented in Table 8. When extrapolated to the entire benchmark, the process requires 7,730,300 input tokens and 1,474,300 output tokens. The total cost amounts to approximately $60 ($38.65 for input tokens plus and $22.11 for output tokens). By implementing batch processing and prompt cache, we were able to reduce this cost by half to nearly $30. While we are actively exploring more cost-effective and efficient evaluation methods, we opted to use the *state-of-the-art* GPT-4o model for evaluation to ensure optimal human alignment and automated assessment quality.

While ISG-Agent does require more computational resources compared to unified models like show-o, as we emphasized in our paper, ISG-Agent serves as an exploration of the upper performance bound for the interleaved generation task, where performance takes precedence over resource efficiency. We appreciate your observation regarding resource usage. As shown in Tables 9 and 10, our analysis of the average token count per query reveals that the time complexity between unified models and ISG-Agent via API is actually comparable.

## E.2  ABLATION STUDY WITH IMAGE INPUT

We explore the vision-dependent of our benchmark, where queries within ISG-BENCH can only be answered when providing image input (Chen et al., 2024c). We evaluated the unified model Show-O and one compositional frameworks (Claude + SD3). Our findings reveal that for Show-O (Table 11), the impact is particularly pronounced at the image and block levels, while showing

Table 11: Vision input enhances performance across all levels except holistic reasoning for Show-o. This suggests the model's capabilities are insufficient to simultaneously process visual inputs and generate images. (w. - With, w.o. - Without)

| Level | Style | Prog. | 3D | Dec. | I-T C. | Temp. | VST | VQA | Avg. |
|---|---|---|---|---|---|---|---|---|---|
| Structure (w.o.) | 0.000 | 0.000 | 0.000 | 0.000 | 0.000 | 0.000 | 0.000 | 0.000 | 0.000 |
| Structure (w.) | 0.000 | **0.027** | **0.510** | **0.080** | **0.269** | **0.500** | **0.087** | **0.733** | **0.276** |
| Image (w.o.) | 0.000 | 0.000 | 0.000 | 0.000 | 0.000 | 0.000 | 0.000 | - | 0.000 |
| Image (w.) | 0.000 | **0.002** | **0.035** | **0.009** | **0.001** | **0.220** | **0.060** | - | **0.047** |
| Block (w.o.) | 1.000 | 1.000 | 1.000 | 1.000 | 1.000 | 1.000 | 1.000 | 1.000 | 1.000 |
| Block (w.) | 1.000 | **1.028** | **1.013** | **1.191** | **1.938** | **3.417** | **1.250** | **3.227** | **1.758** |
| Holistic (w.o.) | **2.305** | **2.721** | 1.732 | **2.780** | 2.140 | 2.530 | **3.360** | 1.787 | **2.419** |
| Holistic (w.) | 1.696 | 2.233 | 1.397 | **2.728** | 1.866 | **2.480** | 3.019 | 1.945 | 2.170 |

Table 12: Vision input enhances performance across all levels for Claude 3.5 + Stable Diffusion 3. (w. - With, w.o. - Without)

| Level | Style | Prog. | 3D | Dec. | I-T C. | Temp. | VST | VQA | Avg. |
|---|---|---|---|---|---|---|---|---|---|
| Structure (w.o.) | **0.005** | 0.000 | **0.500** | 0.090 | 0.080 | 0.330 | 0.000 | 0.247 | 0.156 |
| Structure (w.) | 0.000 | 0.000 | 0.030 | **0.760** | **0.313** | **0.500** | 0.000 | **0.980** | **0.323** |
| Image (w.o.) | **0.001** | 0.000 | 0.016 | 0.026 | 0.000 | 0.126 | 0.000 | - | 0.024 |
| Image (w.) | 0.000 | 0.000 | **0.027** | **0.484** | 0.000 | **0.302** | 0.000 | - | **0.116** |
| Block (w.o.) | **1.019** | 1.000 | 1.013 | 1.219 | 1.327 | 1.900 | 1.000 | 2.250 | 1.341 |
| Block (w.) | 1.000 | 1.000 | **1.048** | **4.904** | **3.380** | **3.357** | 1.000 | **8.011** | **2.962** |
| Overall (w.o.) | 3.640 | 4.877 | 1.848 | 1.850 | 5.353 | 3.290 | 5.469 | 1.827 | 3.519 |
| Overall (w.) | **5.179** | **6.435** | **3.874** | **7.306** | **7.912** | **5.290** | **6.168** | **7.865** | **6.254** |

negative effects at the holistic level, which we attribute to the model's suboptimal performance. As for Claude + SD3 (Table 12), the presence or absence of images significantly impacts all four levels in the compositional frameworks.

Hallucination in Show-o

**Task**: Realistic Object Detection
**Instruction**: Select and extract [left cat, right cat, fan, toy] from the image. For each object, generate a separate and independent image for each object that closely resembles its state. Display the object image followed by its detailed caption according to the sequence of the previous list. Output Requirement: Start with the whole image description. Then, for each object, display the object's image following its caption. For example: [Whole image Description]. <Object 1 image> [Object 1 caption]. <Object 2 image> [Object 2 caption]. <Object 3 image> [Object 3 caption]....

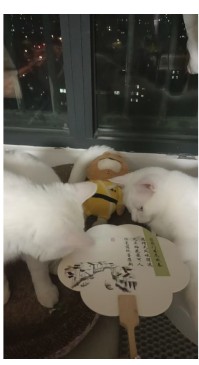

**Model Output**:

Figure 39: Hallucination in Show-o.

## F CASE STUDY

We provide examples of each task in this Section.

For Visual Story Telling, please refer to Figure 40, Figure 41 and Figure 42. For VQA with Image Generation, examples are shown in Figure 43 and Figure 44. For Temporal Prediction, please refer to Figure 45 and Figure 46. Examples of Image-text Complementation are shown in Figure 47 and Figure 48. For Style Transfer, examples are shown in Figure 49, Figure 50, Figure 51 and Figure 52. For Image Decomposition, examples are shown in Figure 53, Figure 54 and Figure 55. Examples of 3D Transformation are shown in Figure 56 and Figure 57. Examples of Progressive Image Transformation are shown in Figure 58, Figure 59 and Figure 60.

Example of Image-based visual storytelling

**Input**: Based on <**image1**>, continue the story, presenting it through 4 new PNG images with descriptions. Ensure each text includes connecting phrases for smooth transitions. For example, {*1st image*} *[description1]*, {*2nd image*} *[description2]*, {*3rd image*} *[description3]*.

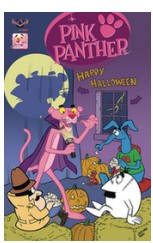

**Golden answer:**

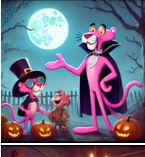

<**image1**> On a moonlit Halloween night, the Pink Panther, dressed as a vampire, leads his friends into a spooky adventure. The eerie glow of the full moon sets the perfect tone for their daring quest.

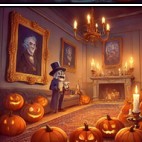

<**image2**> Their journey leads them to a mysterious mansion, filled with carved pumpkins and flickering candles. The Pink Panther cautiously enters, guiding his friends into the dimly lit room where shadows dance on the walls.

<**image3**> Deeper inside the mansion, the group discovers a hidden treasure chest. With excitement, the Pink Panther opens it to reveal sparkling jewels and gold, their eyes widening at the unexpected fortune.

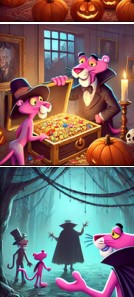

<**image4**> Just as they celebrate their find, ghostly figures emerge from the shadows, and the mansion begins to shake. The Pink Panther quickly devises a plan, using his cape to distract the spirits, allowing the group to make a swift escape into the night.

Figure 40: Example of Image-based visual storytelling.

---

**Example of Text-based visual storytelling**

**Input**: I will tell you the captions of three pictures of this story: 1. The squirrels in the field of poppies are musicians, playing the piano and singing in harmony, as indicated by the musical note and piano in the background. 2. A musical instrument with a story.3.Twelve young squirrels perform a dance routine on stage, their movements fluid and in harmony. Based on the text, continue the story, presenting it through 5 new PNG images with descriptions. Ensure each text includes connecting phrases for smooth transitions. For example, {*1st image*} *[description1],* {*2nd image*} *[description2],* {*3rd image*} *[description3].*

- - - - - - - - - - - - - - - - - - - - - - - - - - - - - - - - - - - - - - - -

**Golden answer:**

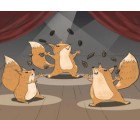 <**image1**> On a brightly lit stage, squirrels enthusiastically juggle acorns, captivating the audience with their synchronized routine. Their movements are smooth, and the spotlight highlights their joyful performance, setting a lively tone for the rest of the show.

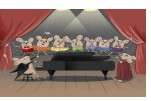 <**image2**> In the next act, a group of mice gather around a grand piano. A mouse in a red dress stands proudly while others sing in unison. Their melodic voices fill the theater, each mouse playing a unique part in creating a harmonious ensemble.

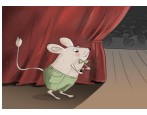 <**image3**> As the curtains draw back for the next scene, a shy mouse peeks from behind them. His nervous yet determined expression hints at his upcoming solo performance, adding suspense and excitement to the audience's anticipation.

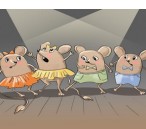 <**image4**> The lights dim, and a quartet of mice in colorful outfits take center stage, showcasing a well-rehearsed dance routine. They twirl and leap, their tails swirling in rhythm, and the audience can't help but be enchanted by their delightful performance.

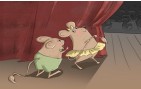 <**image5**> Backstage, two mice prepare for their big moment, anxiously awaiting their turn to shine. Their excitement is palpable as they practice their steps one last time, knowing that their grand finale will bring the entire show to a stunning conclusion.

Figure 41: Example of Text-based visual storytelling

> **Example of Image & text-based visual storytelling**
>
> **Input**: I'll tell you what happens next in the story: A robot and a button will go on an adventure. Based on <**image1**> and the provided text, continue the story, presenting it through 6 new PNG images with descriptions. Ensure each text includes connecting phrases for smooth transitions. For example, {*1st image*} *[description1], {2nd image} [description2], {3rd image} [description3].* 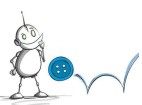
>
> ---------------------------------------------------------------
>
> **Golden answer:**
>
> 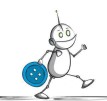 <**image1**> The little robot is happily marching forward, holding a large blue button as if it's his prized possession.
>
> 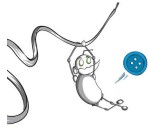 <**image2**> Suddenly, while swinging from a wire, the button slips from his hand and flies into the air. The robot looks startled.
>
> 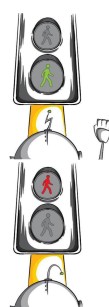 <**image3**> The robot stands underneath a pedestrian traffic light, eagerly waiting for the green light to guide him forward.
>
> <**image4**> The light turns red, and the robot looks disappointed as he realizes it's not time to cross yet.
>
> 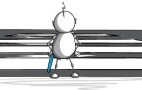 <**image5**> As the robot prepares to step forward, he stares at the road ahead, still determined to retrieve his lost button.
>
> 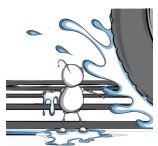 <**image6**> A huge splash of water from a nearby passing vehicle drenches the robot, leaving him soaking wet as he stands on the crosswalk.

Figure 42: Example of Image & text-based visual storytelling

---

**Object Q&A and Explanation**

**Input**: What device is depicted in the picture <**image1**>? Provide detailed information to demonstrate its primary function, historical evolution, and the key steps to use it properly or any other relevant facts. You should answer my question in text first and then generate four interleaved images and text to thoroughly explain the device.

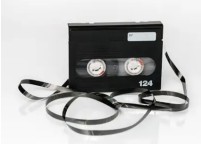

---

**Golden answer:**

The device depicted in the picture is a cassette tape, an analog magnetic tape format used for audio recording and playback. First introduced in the early 1960s by Philips, the cassette tape revolutionized personal audio by offering a compact, portable, and easy-to-use medium. Its primary function is to store sound data, which can be played back using a cassette player or recorder.

The cassette contains two reels of magnetic tape that move across a read-write head, which translates the recorded audio into sound waves. <**image1**>

To use a cassette tape, insert it into a compatible player, press play, and the tape will move through the device, producing sound. <**image2**>

Over time, cassette tapes were replaced by digital formats, but they remain a symbol of the 1980s and 1990s music culture, offering a sense of nostalgia. <**image3**>

While now largely obsolete, cassette tapes are still appreciated by collectors and audiophiles for their analog sound and the physical act of rewinding and flipping the tape. <**image4**>

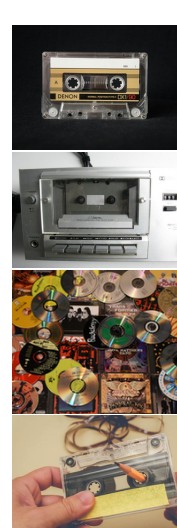

Figure 43: Object Q&A and Explanation.

---

**Example of Historical Event/Artifact Analysis**

**Input**: What is the historical significance of the site or monument depicted in the picture <**image1**>? Provide detailed information about its history, including when it was built, its cultural importance, and any relevant events associated with it. You should answer my question in text first and then generate four interleaved images and text to thoroughly explain the site or monument.

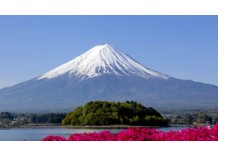

---

**Golden answer:**

The site depicted in the picture is Mount Fuji, Japan's highest peak and one of the country's most iconic natural landmarks. Standing at 3,776 meters (12,389 feet) tall, Mount Fuji is a stratovolcano that last erupted in 1707 during the Edo period.

The mountain's nearly symmetrical cone shape makes it a prominent feature of the Japanese landscape and a symbol of beauty and tranquility. <**image1**>

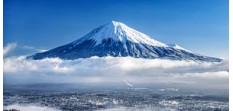

Mount Fuji holds profound cultural and spiritual significance in Japan. It is considered a sacred site in Shinto, the indigenous religion of Japan, and has been a subject of veneration and pilgrimage for centuries. The mountain is frequently depicted in Japanese art and literature, including famous woodblock prints by Katsushika Hokusai and Utagawa Hiroshige. <**image2**>

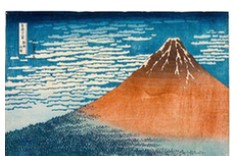

The mountain is also a popular destination for climbers and tourists. The official climbing season is from early July to early September, and thousands of people ascend the mountain each year to experience its scenic beauty and spiritual significance. <**image3**>

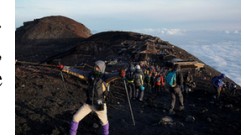

As a UNESCO World Heritage site, Mount Fuji is celebrated for its natural beauty and its role in Japanese culture and history, making it a symbol of national pride and a focal point of both natural and cultural heritage. <**image4**>

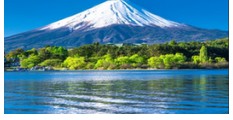

Figure 44: Example of Historical Event/Artifact Analysis.

---

**Example of Real World Simulation**

**Input**: I will give you a picture of a vegetable spiralizer cutting a squash into spirals <image1>. Please use a combination of 4 images and text to show what will happen to this squash. For example, [whole description] {1st image}, {2nd image}, {3rd image}.

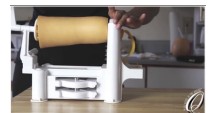

---

**Golden answer:**

The vegetable spiralizer begins cutting the squash into long spirals. As the handle is turned, the squash is pushed against the blade, creating thin spirals. The squash continues to spiralize, and the length of the squash gradually reduces. The spirals form a continuous, long string. <image1> <image2> <image3> <image4>

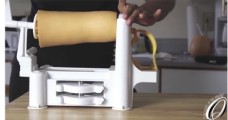 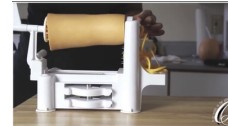 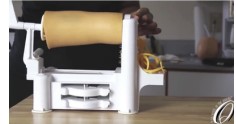 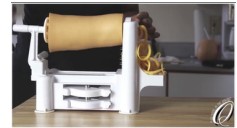

Figure 45: Example of Real World Simulation.

---

**Example of Painting Process Generation**

**Input**: I will give you a painting <image1>. Please tell me how to draw this image in 5 steps using a combination of images and text. For example, *{1st image} [description1], {2nd image} [description2], {3rd image} [description3].*

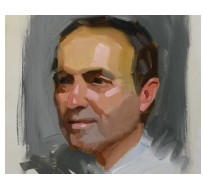

- - - - - - - - - - - - - - - - - - - - - - - - - - - - - - - - - - - - - - - -

**Golden answer:**

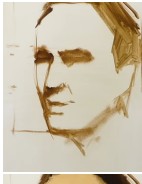

<**image1**> Step 1: Start with an underpainting or basic sketch to block in the shapes and form of the face.

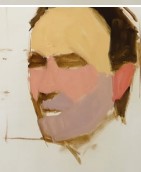

<**image2**> Step 2: Add base skin tones and block in large areas of light and shadow.

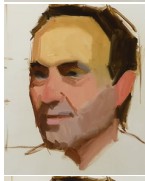

<**image3**> Step 3: Gradually refine the shapes and add details to features like the eyes, nose, and mouth.

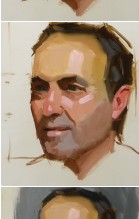

<**image4**> Step 4: Work on smoothing transitions and adding further detail, focusing on depth.

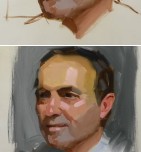

<**image5**> Step 5: Finalize the painting by refining edges, adding highlights, and introducing a subtle background.

Figure 46: Example of Painting Process Generation.

Example of HowTo

**Input**: Please tell me 5 steps on How to shave. Generate an image for each step, and write a brief description after each image. For example, {*1st image*} *[description1]*, {*2nd image*} *[description2]*, {*3rd image*} *[description3]*.

- - - - - - - - - - - - - - - - - - - - - - - - - - - - - - - - - - - - - - - - - - -

**Golden answer:**

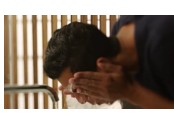 <**image1**> First, keeping your skin moisturized can effectively reduce cuts, nicks, and irritation during shaving. We recommend shaving immediately after a shower or while in the shower. Keeping the skin moist softens your facial hair, allowing the razor to glide more smoothly.

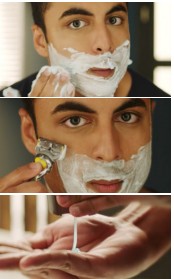 <**image2**> Apply shaving cream or shaving foam - lather it up and apply evenly to moisturize the hair, allowing the razor to glide smoothly and effectively reducing irritation, such as razor burn, during shaving.

<**image3**> Gently glide the razor across your face, softly removing the bread. Rinse your face with clean water and pat it dry.

<**image4**> Apply an aftershave moisturizer to replenish moisture and soothe the skin.

<**image5**> You finished shaving.

Figure 47: Example of HowTo.

---

**Example of Scientific Phenomenon Explanation**

**Input**: How is the phenomenon in the picture <**image1**> scientifically explained? Provide a detailed explanation of the principles behind the phenomenon, including relevant knowledge. You should answer my question in text first and then generate four interleaved images and text to thoroughly explain the phenomenon. 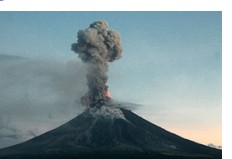

---

**Golden answer:**

The phenomenon in the picture is a volcanic eruption, which occurs when magma from beneath the Earth's crust is expelled to the surface.

This eruption is often characterized by the expulsion of lava, ash, and gases. <**image1**>

The magma originates from the mantle, where it melts due to high pressure and temperature. As magma rises through the Earth's crust, it collects in a magma chamber beneath the volcano. When pressure in the chamber exceeds the strength of the surrounding rock, an eruption occurs. <**image2**>

During the eruption, lava flows from the volcano, while volcanic ash and gases are ejected into the atmosphere. The composition of lava can vary, influencing the eruption style, with basaltic lava producing relatively gentle eruptions and more viscous andesitic or rhyolitic lavas leading to explosive eruptions. <**image3**>

Volcanic eruptions can also create new landforms such as volcanic cones or calderas, shaped by the accumulation of erupted materials. <**image4**>

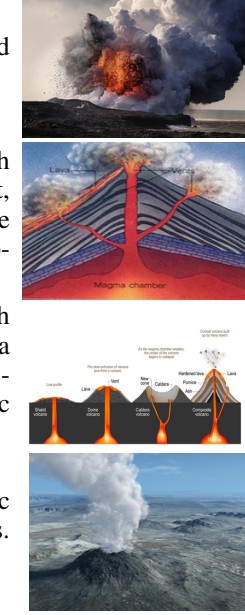

Figure 48: Example of Scientific Phenomenon Explanation.

Example of Art Style Transfer

**Input**: I will give you an image <**image1**>. Using this image, create 3 versions of this image in 3 different artistic styles in order: Dreamweave, Dapple, and Watercolor. Focus on transforming the whole image style while maintaining the subject's features. For each image, provide a brief description of the style. Descriptions should be put after each image. For example, {*1st image*} *[style1]*, {*2nd image*} *[style2]*, {*3rd image*} *[style3]*.

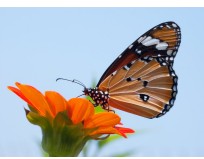

- - - - - - - - - - - - - - - - - - - - - - - - - - - - - - - - -

**Golden answer:**

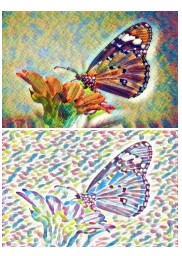

<**image1**> is in Dreamweave style, characterized by a surreal, vibrant, and slightly distorted color pattern that blends colors fluidly, creating an almost dream-like effect.

<**image2**> is in Dapple style, which features a pointillism-like texture with overlapping dots or small patches of color that create a textured and dynamic look.

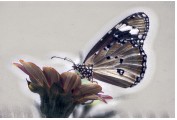

<**image3**> is in Watercolor style, distinguished by soft, flowing colors with blended edges that mimic the translucent and layered appearance typical of watercolor paintings.

Figure 49: Example of Art Style Transfer.

Example of Scene Attribute Transfer

**Input**: Given a photograph <**image1**>, generate 4 images that transform this scene across 4 following conditions in order: foggy, dusk, cloudy, night. Please provide a brief text description explaining the changes made. Descriptions should be put after each image. For example, {*1st image*} *[description1]*, {*2nd image*} *[description2]*, {*3rd image*} *[description3]*.

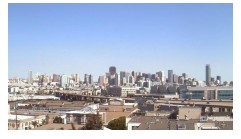

- - - - - - - - - - - - - - - - - - - - - - - - - - - - - - - - -

**Golden answer:**

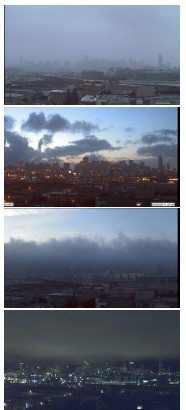

<**image1**> shows a fog-covered city with reduced visibility and dimmed lighting compared to the clear, sunny condition in the original image.

<**image2**> shows a twilight scene with the city lights starting to illuminate and a darker sky.

<**image3**> displays a partially overcast sky with clouds covering much of the skyline, creating a gloomy ambiance.

<**image4**> features a fully dark environment with city lights prominently visible, contrasting with the bright daytime in the original image.

Figure 50: Example of Scene Attribute Transfer.

---

**Example of Photo Variation**

**Input**: Given a photograph <image1>, create 4 new images by applying the following adjustments in order: reduced brightness and exposure; reduced brightness and increased contrast; decreased brightness; greener tone. Please provide a brief text description explaining the changes made. Descriptions should be put after each image. For example, {*1st image*} *[description1]*, {*2nd image*} *[description2]*, {*3rd image*} *[description3]*. 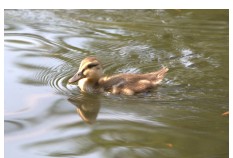

- - - - - - - - - - - - - - - - - - - - - - - - - - - - - - - - - - - - - - -

**Golden answer:**

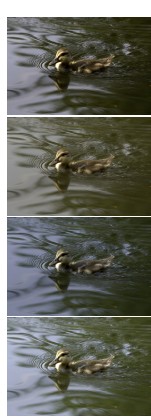

<**image1**> has reduced brightness and exposure, making the overall scene darker.

<**image2**> has reduced brightness and increased contrast, giving it a more pronounced difference between light and dark areas.

<**image3**> has decreased brightness, resulting in a more subdued and dim appearance.

<**image4**> has a greener tone, altering the color balance to emphasize green hues in the scene.

Figure 51: Example of Photo Variation.

Example of Portrait Variation

**Input**: Given input portrait photograph <image1>, create 4 new images by applying the following adjustments in order: lightly better lighting and contrast; more serious expression and a slightly different angle; with eyes closed or looking down; a slightly different facial expression and head tilt. Please provide a brief text description explaining the changes made. Descriptions should be put after each image. For example, {*1st image*} *[description1],* {*2nd image*} *[description2],* {*3rd image*} *[description3].*

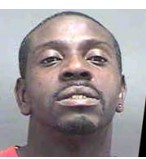

**Golden answer:**

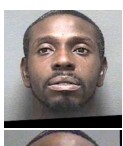 <image1> appears to have slightly better lighting and contrast.

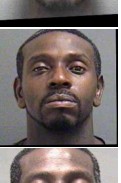 <image2> shows the subject with a more serious expression and a slightly different angle.

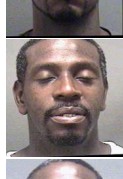 <image3> depicts the subject with his eyes closed or looking down.

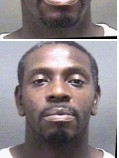 <image4> is with a slightly different facial expression and head tilt.

Figure 52: Example of Portrait Variation.

---

Example of Realistic Image Decomposition

**Input**: Select and extract grenade, flashlight, bullet, dog-tag, compass, badge from the image <**image1**>. If capable, Segment the object directly from the image, else generate a similar one. Output the object image and its detailed caption according to the sequence of previous list. Output Requirement: Start with the whole image description. Then, for each object, display the object's image following its caption. When multiple objects interact, describe them together with conjunctions. For example: *[Whole image description] {1st object's image} [description1], {2nd object's image} [description2], {3rd object's image} [description3].*

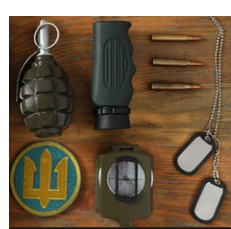

---

**Golden answer:** A collection of military objects laid out on a wooden surface.

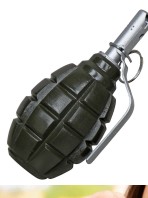 A dark green hand grenade <**image2**> with a metallic safety pin is placed in the upper left corner.

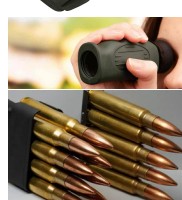 A black and green flashlight grip <**image3**> with a textured surface lies beside the grenade.

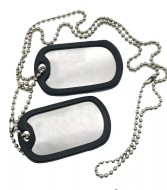 Three brass bullets <**image4**> are lined up on the right, each with a copper tip.

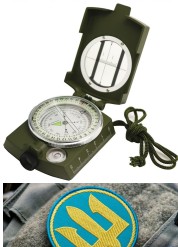 Two dog tags <**image5**> on a metal chain are positioned in the upper right corner.

A green military compass <**image6**> with a transparent cover is displayed near the center.

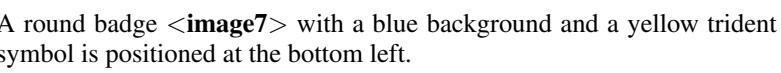 A round badge <**image7**> with a blue background and a yellow trident symbol is positioned at the bottom left.

Figure 53: Example of Realistic Image Decomposition.

---

### Example of Synthetic Image Decomposition

**Input**: Select and extract bread, milk, egg, rice from the image <**image1**>. If capable, Segment the object directly from the image, else generate a similar one. Output the object image and its detailed caption according to the sequence of previous list. Output Requirement: Start with the whole image description. Then, for each object, display the object's image following its caption. When multiple objects interact, describe them together with conjunctions. For example: *[Whole image description] {1st object's image} [description1], {2nd object's image} [description2], {3rd object's image} [description3].*

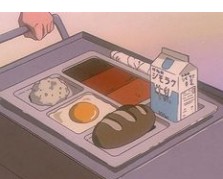

---

**Golden answer:** A simple meal tray featuring bread, milk, egg, and rice.

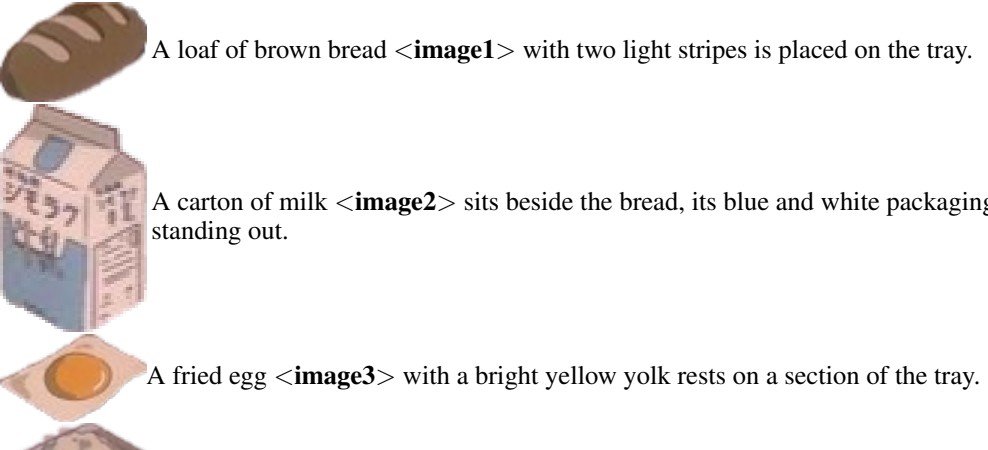

A loaf of brown bread <**image1**> with two light stripes is placed on the tray.

A carton of milk <**image2**> sits beside the bread, its blue and white packaging standing out.

A fried egg <**image3**> with a bright yellow yolk rests on a section of the tray.

A ball of white rice <**image4**> is placed next to the egg, completing the meal.

Figure 54: Example of Synthetic Image Decomposition.

---

**Example of Semantic Decomposition**

**Input**: Decompose the image <**image1**> into left-view, mid-view, right-view regions base on the image composition. If capable, Segment the region directly from the image, else generate a similar one. Display the generated region image and its detailed caption according to the sequence of previous list. Output Requirement: Start with the whole image description. Then, for each region, display the object's image following its caption. For example: *[Whole image description] {1st region's image} [description1], {2nd region's image} [description2], {3rd region's image} [description3].*

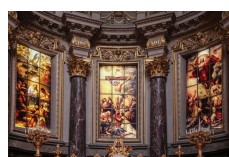

- - - - - - - - - - - - - - - - - - - - - - - - - - - - - - - - - - - - - - -

**Golden answer:** This image showcases a church interior, richly decorated with religious art.

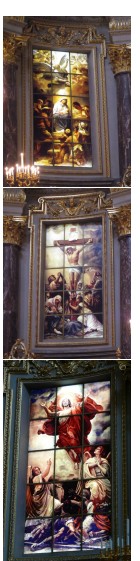

The left view <**image1**> features a painting that illustrates the Baptism of Christ, with angels in attendance, set against a backdrop of dramatic clouds and golden light, framed by dark marbled columns.

The mid-view <**image2**> is dominated by the Crucifixion scene, where Christ is centered on the cross, flanked by sorrowful figures under a stormy sky, emphasizing the solemnity of the moment.

The right view <**image3**> depicts the Ascension of Christ, with apostles gazing upwards as Christ rises, surrounded by a halo of light, conveying a sense of divine transcendence.

Figure 55: Example of Semantic Decomposition.

---

**Example of Multi-view Scene Generation**

**Input**: Here is an image <**image1**> of an object. based on this image, create a series of 4 images showing views from following perspectives in order : right 30, right 15, left 45, left 60. For each image, provide a brief description of the angle. Descriptions should be put after each image. Your response should follow this structure: *30 degrees right: {1st image}, 15 degrees right: {2nd image}, 45 degrees left: {3rd image}, 60 degrees left: {4th image}.*

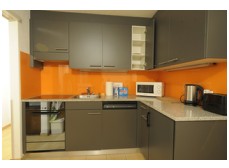

- - - - - - - - - - - - - - - - - - - - - - - - - - - - - - - - - - - - - - - - - - - - - - - - - - - - - - - - - - - - - - - - - - -

**Golden answer:**

The perspective of the object from 30 degrees right: <**image1**>

The perspective of the object from 15 degrees right: <**image2**>

The perspective of the object from 45 degrees left: <**image3**>

The perspective of the object from 60 degrees left: <**image4**>

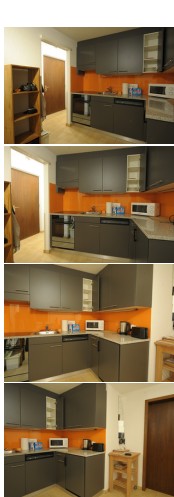

Figure 56: Example of Multi-view Scene Generation.

Example of Multi-angle Object Generation

**Input**: Here is an image <**image1**> of an object. Use this image as the reference angle and generate four additional images of the object from the following angles: 60 degrees left, 30 degrees left, 30 degrees right, and 60 degrees right. Your response should follow this structure: *60 degrees left: {1st image}, 30 degrees left: {2nd image}, 30 degrees right: {3rd image}, 60 degrees right: {4th image}.*

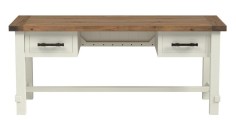

- - - - - - - - - - - - - - - - - - - - - - - - - - - - - - - - - - - -

**Golden answer:**

The perspective of the object from 60 degrees left: <**image1**>

The perspective of the object from 30 degrees left: <**image2**>

The perspective of the object from 30 degrees right: <**image3**>

The perspective of the object from 60 degrees right: <**image4**>

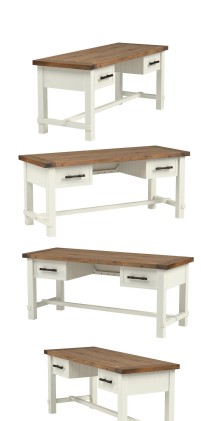

Figure 57: Example of Multi-angle Object Generation

---

**Example of Text-guided Animation**

**Input**: Increase the number of pancake layers from <**image1**>, making the stack taller. Reduce the amount of toppings on the pancakes, keeping the overall composition the same but with a more minimal topping arrangement. <**image1**> represents the initial state, and the provided text describes the changes needed. Create a series of 5 PNG images that gradually transition from the initial state to the final state. Please provide a brief text description explaining the changes made. Descriptions should be put after each image. For example: {*second stage*} [*description1*], ... {*final stage*} [*description5*]. Ensure that the transitions are natural and progressively illustrate the changes between states.

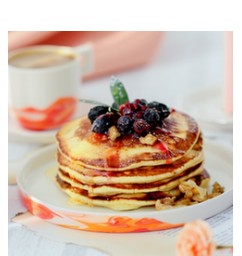

- - - - - - - - - - - - - - - - - - - - - - - - - - - - - - - - - -

**Golden answer:**

At the second stage <**image1**>, the stack of pancakes appears slightly taller with an added layer. The toppings remain similar, but the overall arrangement looks a bit more minimal, maintaining the berries and syrup.

In the third stage <**image2**>, the pancake stack continues to grow taller with another added layer. The toppings have been reduced further, with fewer berries on top, enhancing the minimalistic appearance.

The fourth stage <**image3**> shows an even taller stack with another layer added. The quantity of berries decreases slightly, keeping the minimal topping arrangement consistent.

At the fifth stage <**image4**>, the pancake stack continues to rise with an additional layer. The berries appear to be fewer, maintaining a minimal topping display.

At the final stage <**image5**>, the pancake stack reaches its maximum height with the tallest stack yet. The toppings are further minimized, with just a few berries and a hint of syrup, achieving the intended minimalistic composition.

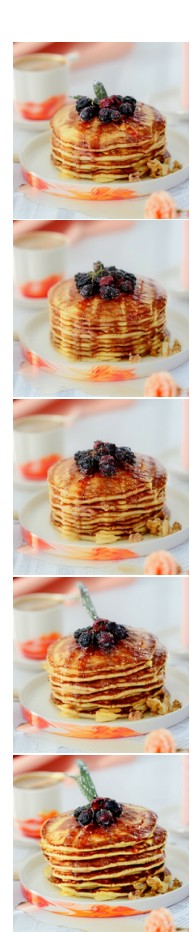

Figure 58: Example of Text-guided Animation

---

Example of Image-guided Animation

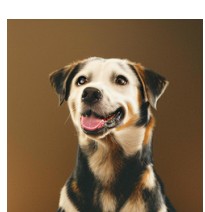 **Input**:<**image1**> (left) represents the initial state, and <**image2**> (right) represents the final desired state. Create a series of 4 PNG images that gradually morph from the initial state to the final state. Include a brief text description after each image explaining the changes made in each step. For example: {*second stage*} *[description1]. ...* {*penultimate stage*} *[description4]..* Ensure that the transitions are natural and progressively illustrate the changes between states. 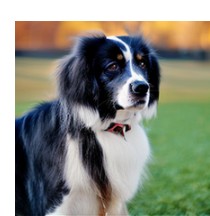

- - - - - - - - - - - - - - - - - - - - - - - - - - - - - - - - - - - - - - - -

**Golden answer:**

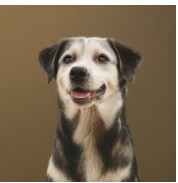 At the second stage <**image1**>, the fur appears slightly longer and the color pattern remains consistent, though the overall form of the dog seems slightly larger and the background remains unchanged.

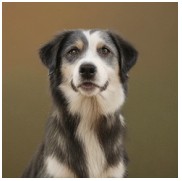 In the third stage <**image2**>, the dog's ears appear to be lengthening slightly, and subtle changes in the shading on the fur give it a slightly fuller appearance, while the background maintains its consistency.

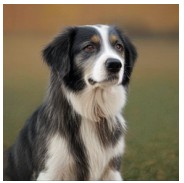 In the fourth stage <**image3**>, there is a noticeable increase in the length and volume of the dog's fur, with a more defined transition in coloration on the coat. The background starts to blend more with the dog's fur tones, creating a more cohesive look.

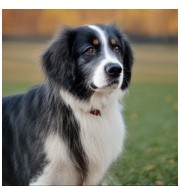 At the penultimate stage <**image4**>, the dog's fur has grown significantly longer and thicker, resembling more closely that of a Border Collie. The overall composition of colors on the coat reflects this transformation, and the background has transitioned to an outdoor scene, enhancing the natural look of the dog.

Figure 59: Example of Image-guided Animation

---

**Example of Attribute-guided Image Generation**

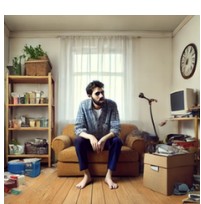

**Input**: I will give you an image of a man sitting in a room <**image1**>. Please show me the process of the man becoming increasingly poor, using a combination of 4 images and text. For example, {*1st image*} *[description1], {2nd image} [description2], {3rd image} [description3]*.

- - - - - - - - - - - - - - - - - - - - - - - - - - - - - - - - - - - - - - - - - -

**Golden answer:**

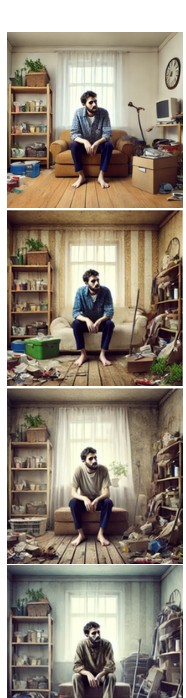

<**image1**> The man is sitting in a room that appears somewhat cluttered but still functional. His clothes are clean, and the room, although slightly messy, is in relatively good condition

<**image2**> The man's clothes appear worn, and the room is becoming more disorganized, with more clutter accumulating around him.

<**image3**> The man is visibly more disheveled, and the room has significantly deteriorated, with trash and debris scattered across the floor.

<**image4**> The man looks exhausted, his clothes are tattered, and the room is in a state of disrepair, filled with broken furniture and debris, reflecting deep poverty.

Figure 60: Example of Attribute-guided Image Generation

