# OpenReview forum: "Interleaved Scene Graphs for Interleaved Text-and-Image Generation Assessment"
_ICLR.cc/2025/Conference — ICLR 2025 Spotlight_

### Official Review · Reviewer_D3cy · 2024-11-02

**Soundness:** 3
**Presentation:** 3
**Contribution:** 2
**Rating:** 8
**Confidence:** 4

**Summary:**

The paper presents a new evaluation method and benchmark for the interleaved image-text generation task, which aims to generate coherent image text contents for queried use cases, such as instruction generation, visual storytelling, and others (Table 2, Figure 3). For evaluation, the study presents converting multimodal queries into atomic questions for a better scoring than MLLM-based evaluations. The constructed benchmark contains 1150 samples covering 21 generation scenarios. In addition to existing methods, an agent framework named ISG-AGENT is also proposed to enhance the performance.

**Strengths:**

1. The proposed evaluation aims to address the problem that MLLM evaluation alone is not reliable. This is a valid question, and the proposed improvements: (1) multi-granular evaluation; (2) converting to question answering, do serve as an effective add-on for evaluation.

2. The benchmark covers a wide range of evaluation scenarios for the interleaved image-text generation.

3. The paper is presented well, with sufficient discussion on evaluation pipeline, related works, and evaluation categories.

**Weaknesses:**

1. One concern is on the definition of the task “interleaved image-text generation,” related to the discussion in the first paragraph of introduction. The current definition of the task is based on the application scenarios, which are valid and of good application value. However, it fails to provide extra insights to the task, including what are the unique challenges, and how to balance the categories and samples in the evaluation benchmark.

For the categories and samples percentage, the benchmark already covers a wide range of scenarios (Table 2, Figure 3). But the motivation of the percentage and construction remains unclear: why certain scenarios are selected, how to decide between 50/100 samples, and whether there is an “ideal” distribution to follow for the task.

2. The experiment evaluates various interleaved generative frameworks.

Related to the task’s unique challenge discussions, it would be helpful to discuss whether the image and text generation can be disentangled: providing insights on whether agent systems or naturally multimodal (unified) models (e.g., GPT4o and others) are more promising in the future. The experiments discuss the performance of existing models in these two lines, but fail to suggest future directions on them.

The current observation is that the open-sourced “unified model” is much worse compared with agent systems. This is partially because some of the better unified models are not open-sourced. It would be interesting to add a few examples compared with the demos shown in their paper/webpage, to provide insights on “future better unified models.” (e.g., https://openai.com/index/hello-gpt-4o/ Explorations of capabilities: Visual narratives - Sally the mailwoman) And when both directions have comparable base capacity, what are there relative strengths and weaknesses.

3. The “GT” in abstract and Table 1 still works on pre-selected evaluated models, instead of extending to arbitrary new models to evaluate. Therefore, the evaluation could still remain noisy similar to MLLM-scoring.

**Questions:**

1. In Table 1, for OpenLeaf (An et al., 2023), why is the number of samples 30 instead of 660 listed in the paper?

2. It would be helpful to discuss the plan on setting up and maintaining the automatic evaluation benchmark for easier and wider use.

3. More examples on the evaluation questions, and category examples are helpful.

---

> ### Author Response · Authors · 2024-11-19
> **Rebuttal by Authors**
>
> Thank you very much for your valuable feedback. We apologize for any confusion caused by certain details in the paper. We will address each of your concerns and provide explanations to help you better understand the contributions of this paper step by step:
>
> ---
> **Q1.1:** It fails to provide extra insights to the interleaved generation task, including what are the unique challenges.
>
> **A1.1:** Thank you for this valuable suggestion. Since our paper primarily focuses on evaluating interleaved generation rather than proposing new interleaved generation methods, we had not extensively discussed this aspect in the previous version. However, we have now added supplementary discussions about interleaved generation in both the Related Work section and Appendix A to address this concern. In the revised version of our manuscript, we also expand the introduction of the interleaved generation task in the first paragraph of the Introduction section.
>
> ---
> **Q1.2:** For the categories and samples percentage, the benchmark already covers a wide range of scenarios (Table 2, Figure 3). But the motivation of the percentage and construction remains unclear: why certain scenarios are selected.
>
> **A1.2:** Regarding our task selection criteria, we focus on tasks that unified models can perform but other frameworks either cannot handle or perform poorly on. Some of these tasks, such as Visual Story Telling and Instruction Videos (Howto) that you mentioned, are specifically highlighted in the unified model paper. To address the task categorization, we have added a classification tree in the Appendix A to systematically categorize new tasks. This taxonomy will facilitate the evaluation of future tasks using ISG when they are incorporated into the framework.
>
> ---
> **Q1.3:** How to balance the categories and samples in the evaluation benchmark. How to decide between 50/100 samples, and whether there is an “ideal” distribution to follow for the task.
>
> **A1.3:** Thank you for your question. We have set the number of samples for each subcategory to either 50 or 100, aiming to maintain categorical balance in our benchmark design. As for why certain categories contain 100 samples, we follow the reasoning mentioned in our original paper: these tasks were empirically determined to be more common in everyday life.
>
> ---
> **Q2.1:** It would be helpful to discuss whether the image and text generation can be disentangled: providing insights on whether agent systems or naturally multimodal (unified) models (e.g., GPT4o and others) are more promising in the future. The experiments discuss the performance of existing models in these two lines, but fail to suggest future directions on them.
>
> **A2.1:** Thank you for your suggestion. In the revised version, we have expanded the introduction of interleaved generation tasks in the first paragraph of the Introduction section. Additionally, we have included a forward-looking discussion about the prospects of better unified models. Finally, we have enriched our case study with new comparative examples showing the generation results between unified models and compositional frameworks.
>
> ---
> **Q3:** The “GT” in abstract and Table 1 still works on pre-selected evaluated models, instead of extending to arbitrary new models to evaluate. Therefore, the evaluation could still remain noisy similar to MLLM-scoring.
>
> **A3:** Thank you for raising this point. As demonstrated in our paper, MLLM-Scoring can indeed be noisy when ground truth or golden answers are not available. Based on this finding, for future extensions of ISG to new tasks, we recommend collecting human-annotated golden answers, as this significantly improves reliability of MLLM-Scoring.
>
> ---
> **Q4:**  In Table 1, for OpenLeaf (An et al., 2023), why is the number of samples 30 instead of 660 listed in the paper?
>
> **A4:** Thank you for raising this point. In their paper **[1]**, we note that they stated in their introduction: `We collect a benchmark dataset for evaluating open-domain interleaved generation methods, which consists of 30 input queries, covering a wide range of topics and formats.` This confirms that 30 is the correct number.
>
> **[1]** https://arxiv.org/abs/2310.07749 OpenLEAF
>
> ---
> **Q5:** It would be helpful to discuss the plan on setting up and maintaining the automatic evaluation benchmark for easier and wider use.
>
> **A5:** Thank you for your valuable suggestion! In our latest manuscript, we have updated how new tasks can be categorized within our ISG evaluation framework, as shown in Figure 7. We will also provide support for incorporating new tasks when we release our code.
>
> ---
> **Q6:** More examples on the evaluation questions, and category examples are helpful.
>
> **A6:** Thank you for your suggestion. We will include more case studies of unified models in our paper. However, due to space limitations, we plan to showcase additional examples on our forthcoming project website. We appreciate your support!

---

> > ### Comment · Reviewer_D3cy · 2024-11-29
> > **Thank you to the authors for the detailed response.**
> >
> > I will raise my score to an accept, as most of my other concerns have been addressed.

---

> > > ### Author Response · Authors · 2024-11-29
> > > **Thanks!**
> > >
> > > Thank you for raising your score! We sincerely appreciate your time and dedication in reviewing our work and are truly delighted by your strong endorsement of our research.

---

### Official Review · Reviewer_rWir · 2024-11-03

**Soundness:** 3
**Presentation:** 3
**Contribution:** 3
**Rating:** 6
**Confidence:** 3

**Summary:**

the paper proposed one multimodal benchmark, named ISG-BENCH, and one agent-based approach to tackle the problem. the focus is the interleaved image-text data as the model output, which is a promising direction in the field. in the benchmark, there are 4-level evaluation methods, which is structure, block, image, and holistic. these provide multiple perspectives to study the model performance. the proposed agent-based approach also achieves the best among all the existing and the baseline approaches.

**Strengths:**

the proposed benchmark is a multi-granular evaluation set for interleaved text-and-image generation. this provide options to study the model performance through various perspectives.

the proposed benchmark scale is the largest by far, as shown in the paper Tab 1, consisting of 1k+ samples, in 21 categories. this could be the primary contribution to the community.

the agent-based approach utilizes a "Plan-Execute-Refine" structure, with tool use. This compositional approach shows promise in generating high-quality outputs and provides a strong baseline for further research.

**Weaknesses:**

the primary contribution could be the benchmark with multi-granularity evaluation. on one hand, this is great to provide more options. on the other hand, this may suggest this work is a combination of multiple existing approaches, e.g. by combining openleaf and interleavedbench. although the work provides some unique features, e.g. structure-level evaluation, however, the most important evaluation metric could be the holistic evaluation, which has already been studied in the literature.

the multi-granularity evaluation is performed by different approaches. would be it possible to use only one approach (e.g. MLLM) to evaluate all these perspectives in a more unified, simpler, more elegant way?

considering the holistic evaluation may be the most important, it is better to show more study on why the proposed MLLM-as-the-judge is a good judge, e.g, how the evaluation is aligned with the human judgement. if approach A is better than approach B from MLLM-as-the-judge, is it also true for human judgement? what kind of failure cases could be with the MLLM-as-the-judge.

**Questions:**

see weakness.

---

> ### Author Response · Authors · 2024-11-19
> **Rebuttal by Authors**
>
> Thank you very much for your valuable feedback. We apologize for any confusion caused by certain details in the paper. We will address each of your concerns and provide explanations to help you better understand the contributions of this paper step by step:
>
> ---
> **Q1:**  Concerns regarding that ISG is a combination of multiple existing approaches, e.g. by combining openleaf and interleavedbench. And the holistic evaluation has already been studied in the literature.
>
> **A1:** Thank you for your questions. First, we need to clarify that although we all use holistic level evaluation, our approach differs significantly from other papers. Unlike OpenLeaf and InterleavedBench, which rely on models' pretrained knowledge for evaluation without providing standarized golden answer, our method incorporates golden answers, enabling models to make comparisons and produce judgments that better align with human.
>
> Regarding your second point, we need to clarify why multiple evaluation levels are necessary for interleaved instruction following tasks. Consider the example from our paper: `Generate three images following the input image's story, each followed by an image description.` To comprehensively evaluate such outputs:
>
> 1. We need structural evaluation to assess whether the generation follows the correct interleaved structure
> 2. Block-level evaluation is required to verify if the text accurately describes its corresponding image
> 3. Image-level evaluation is necessary to determine whether generated images maintain consistent style and narrative with the input image
>
> Holistic evaluation alone cannot provide such fine-grained assessment across these three levels, nor can it offer interpretable results - it only evaluates the overall quality of the generated image-text content. Therefore, for our tasks, all four levels of evaluation are crucial, and we believe a multi-tiered evaluation framework better presents our assessment results.
>
> ---
> **Q2:** The multi-granularity evaluation is performed by different approaches. would be it possible to use only one approach (e.g. MLLM) to evaluate all these perspectives in a more unified, simpler, more elegant way?
>
> **A2:** Thank you for your valuable suggestion about unifying the four evaluation methods using MLLM. In fact, in our current implementation, we have already unified our approach by using GPT-4o for both VQA tasks and holistic evaluation. However, for structural evaluation, we believe that direct matching through programmatic methods remains more reliable (and also cheaper) than using MLLM. We appreciate your suggestion for considering a unified approach.
>
> ---
> **Q3:**  More study on why the proposed MLLM-as-the-judge is a good judge, e.g, how the evaluation is aligned with the human judgement. if approach A is better than approach B from MLLM-as-the-judge, is it also true for human judgement? what kind of failure cases could be with the MLLM-as-the-judge.
>
> **A3:** Thank you for your suggestion. We have presented the human validation results of MLLM-as-a-Judge in Table 3. Our experimental setup involved sampling from each category to create a validation set of 260 examples for human evaluation. This sample size was chosen considering the lengthy content of interleaved generation and human effort for cross-validation. Our experimental results demonstrate that providing golden answer to MLLM for evaluation yields judgments that align better with human assessments compared to previous methods that rely solely on MLLM's pretrained knowledge. We also add a failure case of MLLM-as-a-Judge to Appendix in our latest manuscript. Thank you for your valuable suggestion.
>
> ---
> Thank you again for taking the time to review our paper. Your feedback has helped us improve both the clarity and quality of our work.

---

### Official Review · Reviewer_4ngT · 2024-11-03

**Soundness:** 4
**Presentation:** 3
**Contribution:** 3
**Rating:** 8
**Confidence:** 3

**Summary:**

This paper presents a unified approach for evaluating interleaved multimodal text-and-image generation using a novel framework called INTERLEAVED SCENE GRAPH (ISG), which assesses generation accuracy at four granular levels through atomic question-based visual verification. Additionally, it introduces ISG-BENCH, a benchmark of 1,150 samples across 21 generation tasks, and ISG-AGENT, a compositional agent framework designed to explore the limits of interleaved generation. The study’s experiments reveal that, while ISG enables precise evaluation aligned with ground truth and human preferences, existing models often fail in accurate instruction-following, particularly in vision-dominated tasks, highlighting areas for improvement in multimodal generation research.

**Strengths:**

- This is a timely work that studies the critical and underexplored challenge of evaluating the recent popular unified models.

- ISG-BENCH provides a comprehensive benchmark with 1,150 carefully designed samples across 21 tasks, supporting standardized evaluation in multimodal generation.

- The paper conducts extensive multi-granular evaluations, offering a thorough assessment that reveals limitations in current models.

- The proposed ISG-AGENT explores the upper bounds of interleaved generation, highlighting strengths and limitations within the framework.

- The paper is very well written and organized.

**Weaknesses:**

- The model evaluation is relatively complex, as many steps involve MLLMs and LLMs to assess results. Given that the author's proposed task is more challenging, how reliable are current MLLMs and LLMs in this evaluation?

- Some tasks seem overly simplistic and not directly relevant to multimodal generation—for instance, Image Decomposition, which resembles a basic computer vision task. Has there been any explanation of the criteria used to filter these task categories?

**Questions:**

- I believe the intent of this benchmark should be to establish samples that are well-suited to tasks requiring both text and image generation for optimal performance, correct? If the agentic approach can yield good results, might this indicate that the benchmark isn't ideally suited for evaluating a unified model? If the benchmark contains some tasks that only a unified model could perform well, that would be even better.

---

> ### Author Response · Authors · 2024-11-19
> **Rebuttal by Authors**
>
> Thank you very much for your valuable feedback. We apologize for any confusion caused by certain details in the paper. We will address each of your concerns and provide explanations to help you better understand the contributions of this paper step by step:
>
> ---
> **Q1:** How reliable are current MLLMs and LLMs in this evaluation?
>
> **A1:** We validated the reliability of MLLMs and LLMs used in our method by comparing their judgments with human-annotated ground truth similar to the setting in MLLM-as-a-Judge **[1]**, as shown in Table 3 of our paper. The results demonstrate high reliability: our method achieved a Pearson similarity score exceeding 0.7, indicating strong correlation with human judgments. Furthermore, for both image-level assessment and question generation tasks, our method attained accuracy scores above 0.8, providing strong evidence for the high reliability of these MLLMs and LLMs.
>
> **[1]** MLLM-as-a-Judge: Assessing Multimodal LLM-as-a-Judge with Vision-Language Benchmark
>
> ---
> **Q2:** Some tasks seem overly simplistic and not directly relevant to multimodal generation—for instance, Image Decomposition, which resembles a basic computer vision task. Has there been any explanation of the criteria used to filter these task categories?
>
> **A2:** Our task selection criteria focused on identifying tasks that unified models can potentially handle well, but other models either cannot address or perform poorly on. For example, the semantic image decomposition task you mentioned, such as segmenting foreground and background from image, requires semantic understanding capabilities that current segmentation models cannot achieve, but unified models show promising potential for.
> While our task selection process was largely empirical, we made efforts to cover all categories we could identify. Furthermore, in our revised manuscript, we have introduced a classification tree to systematically categorize future interleaved generation tasks, which has been updated in the new manuscript's Appendix A.
>
> ---
> **Q3:** Concerns about whether this benchmark is suitable for unified models.
>
> **A3:** Our task selection criteria focus on tasks that unified models are capable of handling, such as Visual Story Telling and style transfer (image editing). While unified models are still in their infancy and their performance remains bad, we believe that a standardized benchmark is crucial for the development of this emerging field. Our benchmark aims to provide a valuable testbed for future unified models.
> We proposed the agentic approach not only to explore the current upper bound of performance and provide insights for future research but also, as discussed in Appendix A, to leverage the agentic framework for generating large-scale interleaved instruction tuning datasets - a direction we are actively pursuing in our follow-up work.
> Our perspective is that while these tasks were originally designed to evaluate unified models (which currently show limited performance), the strong performance of compositional frameworks suggests that both technological approaches - unified and compositional - are valid paths for interleaved generation.
>
> ---
> Thank you again for taking the time to review our paper. Your feedback has helped us improve both the clarity and quality of our work.

---

> > ### Comment · Reviewer_4ngT · 2024-11-26
> >
> > Thanks for the authors' rebuttal. I have no more questions.

---

> > > ### Author Response · Authors · 2024-11-29
> > > **Thank you for your time and effort!**
> > >
> > > We sincerely appreciate your time and dedication in reviewing our work and are truly delighted by your strong endorsement of our research.

---

### Official Review · Reviewer_ASqC · 2024-11-05

**Soundness:** 3
**Presentation:** 3
**Contribution:** 3
**Rating:** 8
**Confidence:** 4

**Summary:**

This paper introduces a new benchmark for evaluating the interleaved text-and-image generation tasks. While previous multimodal benchmarks only focus on evaluating vision understanding ability where only text outputs are required, this benchmark designed 21 text-image generation tasks with 1150 samples in total. The benchmark have multiple-level of evaluation including structure, block, image, and holistic levels. They also propose a ISG-Agent, which utilize external tools to fulfill the requirements of the benchmarks. Results show that existing models fall short of the benchmark and ISO-Agent performs significantly better than baseline models.

**Strengths:**

1. This paper presents first benchmark to evaluate the models that can generate interleaved text-and-image models, which differs fundamentally from previous vision understanding benchmarks like MMMU. The benchmarks feature an important usage of multimodal models for content generation, and can greatly impact the community if well-maintained and updated.
2. The design of ISG-Agent is reasonable and novel, achieving state-of-the-art performance on the ISG-Bench. The introduce of Planning, Tool-usage, and refinement works well in practice.
3.  The analysis part brings many insights such as MLLM-as-a-Judge cannon evaluate well due biases like "image-qualtiy bias", etc.

**Weaknesses:**

1. The evaluation still uses GPT-4o as a judge, which can bring a lot of bias, as section 4.2 states. The
2. It might be a bit unfair to compare ISG-Agent with another model that can generate interleaved text and images in a whole, as the agent still uses external tools and additional inference tokens for planning, etc. I am not saying ISG-Agent is not good. But it should also be compared with other agents for a fair comparison.

**Questions:**

1. What's the cost of evaluating a new model on the ISG-Bench since evaluation heavily relies on the GPT-4 in the process
2. While you recognize the potential bias of using MLLM as a judge for evaluating image quality, did you figure out any ways to mitigate the bias? Have you conducted any human studies to understand how will these kinds of biases affect the evaluation of the benchmark?
3. How many resources does ISG-Bench need to finish a query in the ISG-Bench? Is it comparable with using other models like show-o to directly generate the outputs?
4. There are many samples in your datasets that have images in the query. Have you ever conducted any study that removes the image in the query and keeps only the text part, then conduct the evaluation in a text-input-only setting? This is to see the gap between the normal setting and make sure that these images in the query are really important to generate a good response, which is now a pretty common analysis for the vision understanding benchmarks.

**Details Of Ethics Concerns:**

In section 3.2, the papers say the data of the benchmarks were collected from existing datasets and further annotated by human experts. However, the paper does not mark the source of these datasets and it's a potential concern if there are licence issues of the source datasets.

Besides, as a benchmark, the release of the dataset may also undergo the ethics review process for the safety reason.

---

> ### Author Response · Authors · 2024-11-19
> **Rebuttal by Authors (1)**
>
> Thank you very much for your valuable feedback. We apologize for any confusion caused by certain details in the paper. We will address each of your concerns and provide explanations to help you better understand the contributions of this paper step by step:
>
> ---
> **Q1:** The evaluation still uses GPT-4o as a judge, which can bring a lot of bias, as section 4.2 states.
>
> **A1:** Thank you for your insights. We have enhanced the original MLLM-as-a-Judge setting by providing human-annotated golden answers for comparison, which has improved the alignment with human judgment, achieving a 0.730 agreement score. While we acknowledge that MLLM-as-a-Judge may not be entirely reliable, it has emerged as a specific metric for evaluating T+V → T **[1]** and even T+V → V **[2]** tasks. Although providing ground truth significantly improves its reliability, we recognize this remains an open challenge in the field. We will highlight this limitation in our paper. We appreciate your reminder about this important issue.
>
> **[1]** MLLM-as-a-Judge: Assessing Multimodal LLM-as-a-Judge with Vision-Language Benchmark
>
> **[2]** MJ-Bench: Is Your Multimodal Reward Model Really a Good Judge for Text-to-Image Generation?
>
> ---
>
> **Q2:** It might be a bit unfair to compare ISG-Agent with another model that can generate interleaved text and images in a whole.
>
> **A2:** Thank you for your suggestion. As stated in our paper, ISG-Agent is the first agent framework that leverages external tools for interleaved generation. We indeed found it challenging to identify appropriate baselines for comparison, as interleaved generation is a novel and challenging task. Therefore, as mentioned in our paper, we position ISG-Agent as a baseline framework for this task, and we hope our research will provide valuable insights for future work on developing more advanced models or frameworks for interleaved generation.
>
> ---
>
> **Q3:** What's the cost of evaluating a new model on the ISG-Bench since evaluation heavily relies on the GPT-4 in the process?
>
> **A3:** We conducted a cost analysis on a sample test, with results presented in **Table 1**. When extrapolated to the entire benchmark, the process requires 7,730,300 input tokens and 1,474,300 output tokens. The total cost amounts to approximately 60 dollars (38.65 for input tokens and 22.11 for output tokens). By implementing batch processing and prompt cache, we were able to reduce this cost by half to nearly 30 dollars. While we are actively exploring more cost-effective and efficient evaluation methods, we opted to use the state-of-the-art GPT-4 model for evaluation to ensure optimal human alignment and automated assessment quality.
>
> **Table 1: Average token consumption for 1 sample.**
>
> | Level | Output | Input | Total |
> |--------|------------|--------|--------|
> | Image | 280 | 2,285 | 2,565 |
> | Block | 738 | 3,221 | 3,959 |
> | Holistic | 264 | 1,216 | 1,480 |
> | Overall | 1,282 | 6,722 | 8,004 |
>
> ---
>
> **Q4:** While you recognize the potential bias of using MLLM as a judge for evaluating image quality, did you figure out any ways to mitigate the bias? Have you conducted any human studies to understand how will these kinds of biases affect the evaluation of the benchmark?
>
> **A4:** Thank you for raising this general question. I would like to address it from a broader perspective of MLLM-as-a-Judge, beyond the scope of our current paper. A promising direction for improvement, in my view, would be to break down the evaluation into different rubrics, similar to V-Bench **[3]**. For instance, separating the assessment into correctness and quality metrics could potentially enhance interpretability and reduce bias. We acknowledge that evaluating generative tasks remains inherently challenging. In this work, we conducted human case studies and identified this issue through several specific examples. While a comprehensive quantitative study of this bias would require developing a new testbed, which we are currently working on as a follow-up study, we won't be able to provide quantitative results within the rebuttal period. However, we look forward to sharing our updated findings in future work!
>
> **[3]** VBench : Comprehensive Benchmark Suite for Video Generative Models

---

> > ### Author Response · Authors · 2024-11-19
> > **Rebuttal by Authors (2)**
> >
> > **Q5:** How many resources does ISG-Agent need to finish a query in the ISG-Bench? Is it comparable with using other models like show-o to directly generate the outputs?
> >
> > **A5:** While ISG-Agent does require more computational resources compared to unified models like show-o, as we emphasized in our paper, ISG-Agent serves as an exploration of the upper performance bound for the interleaved generation task, where performance takes precedence over resource efficiency. We appreciate your observation regarding resource usage. As shown in **Tables 2** and **3**, our analysis of the average time consumption between unified models and ISG-Agent via API is actually comparable.
> >
> > **Table 2: Average computing time on A800 servers.**
> >
> > | Category | Anole | Show-o |
> > |----------|-------|--------|
> > | Style Transfer | 43.330 | 120.103 |
> > | Progressive | 47.093 | 91.917 |
> > | 3D Scene | 33.205 | 73.095 |
> > | Image Decomposition | 21.830 | 91.990 |
> > | Image-Text Complementation | 52.280 | 74.335 |
> > | Temporal Prediction | 33.190 | 53.150 |
> > | Visual Story Telling | 28.150 | 111.400 |
> > | VQA | 36.330 | 185.380 |
> >
> > **Table 3: Average tokens and estimate time (600 tokens per second via API) of ISG-Agent in ISG-Bench.**
> >
> > | Category | Overall (avg_input/avg_output) | Time (s) |
> > |----------|-------------------------------|----------|
> > | Style Transfer | 14448/1801 | 27.0 |
> > | Progressive | 14649/2019 | 27.8 |
> > | 3D Scene | 7755/1158 | 14.8 |
> > | Image Decomposition | 12370/1802 | 23.6 |
> > | Image-Text Complementation | 12196/1758 | 23.3 |
> > | Temporal Prediction | 9800/1358 | 18.6 |
> > | VST | 13712/2230 | 26.5 |
> > | VQA | 9128/1139 | 17.1 |
> >
> > ---
> > **Q6:** Have you ever conducted any study that removes the image in the query and keeps only the text part, then conduct the evaluation in a text-input-only setting?
> >
> > **A6:** Thank you for your suggestion, which provides an excellent approach to demonstrate the vision-dependent nature of our benchmark. We evaluated the unified model Show-O and two compositional frameworks (Claude + SD3). Our findings reveal that for Show-O (**Table 4**), the impact is particularly pronounced at the image and block levels while showing negative effects at the holistic level, which we attribute to the model's suboptimal performance. As for Claude + SD3 (**Table 5**), the presence or absence of images significantly impacts all four levels in the compositional frameworks.
> >
> > **Table 4: Vision input enhances performance across all levels except holistic reasoning for Show-o. This suggests the model's capabilities are insufficient to simultaneously process visual inputs and generate images.**
> >
> > | Metric | Style_Transfer | Progressive | 3D_Scene | Image_Decomp. | Image-Text_Comp/ | Temporal_Prediction | VST | VQA | Average |
> > |--------|----------------|-------------|----------|--------------------|-----------------------------|-------------------|---------------------|-----|---------|
> > | Structure (w/o) | 0.000 | 0.000 | 0.000 | 0.000 | 0.000 | 0.000 | 0.000 | 0.000 | 0.000 |
> > | Structure (w/) | 0.000 | **0.027** | **0.510** | **0.080** | **0.269** | **0.500** | **0.087** | **0.733** | **0.276** |
> > | DSG (w/o) | 0.000 | 0.000 | 0.000 | 0.000 | 0.000 | 0.000 |  0.000 | - | 0.000 |
> > | DSG (w/) | 0.000 | **0.002** | **0.035** | **0.009** | **0.001** | **0.220** | **0.060** | - | **0.047** |
> > | TextImGraph (w/o) | 1.000 | 1.000 | 1.000 | 1.000 | 1.000 | 1.000 | 1.000 | 1.000 | 1.000 |
> > | TextImGraph (w/) | 1.000 | **1.028** | **1.013** | **1.191** | **1.938** | **3.417** | **1.250** | **3.227** | **1.758** |
> > | Overall (w/o) | **2.305** | **2.721** | 1.732 | **2.780** | 2.140 | 2.530 | **3.360** | 1.787 | **2.419** |
> > | Overall (w/) | 1.696 | 2.233 | 1.397 | **2.728** | 1.866 | **2.480** | **3.019** | **1.945** | 2.170 |
> >
> > **Table 5: Vision input enhances performance across all levels for Claude 3.5 + Stable Diffusion 3.**
> >
> > | Level | Style_Transfer | Progressive | 3D_Scene | Image_Decomp. | Image-Text_Comp. | Temporal_Prediction | VST | VQA | Average |
> > |--------|----------------|-------------|----------|--------------------|-----------------------------|-------------------|---------------------|-----|---------|
> > | Structure (w.o) | **0.005** | 0.000 | **0.500** | 0.090 | 0.080 | 0.330 | 0.000 | 0.247 | 0.156 |
> > | Structure (w.) | 0.000 | 0.000 | 0.030 | **0.760** | **0.313** | **0.500** | 0.000 | **0.980** | **0.323** |
> > | DSG (w.o) | **0.001** | 0.000 | 0.016 | 0.026 | 0.000 | 0.126 | 0.000 | - | 0.024 |
> > | DSG (w.) | 0.000 | 0.000 | **0.027** | **0.484** | 0.000 | **0.302** | 0.000 | - | **0.116** |
> > | TextImGraph (w.o) | **1.019** | 1.000 | 1.013 | 1.219 | 1.327 | 1.900 | 1.000 | 2.250 | 1.341 |
> > | TextImGraph (w.) | 1.000 | 1.000 | **1.048** | **4.904** | **3.380** | **3.357** | 1.000 | **8.011** | **2.962** |
> > | Overall (w.o) | 3.640 | 4.877 | 1.848 | 1.850 | 5.353 | 3.290 | 5.469 | 1.827 | 3.519 |
> > | Overall (w.) | **5.179** | **6.435** | **3.874** | **7.306** | **7.912** | **5.290** | **6.168** | **7.865** | **6.254** |

---

> > > ### Author Response · Authors · 2024-11-19
> > > **Rebuttal by Authors (3)**
> > >
> > > **Q7:** Concerns of copyright and safety reasons.
> > >
> > > **A7:** Thank you for your comments. We have provided detailed explanations about our datasets and sampling procedures in Appendix B. Regarding licensing, we will align our dataset license with the source datasets' requirements and release our benchmark under the CC 4.0 license, which restricts usage to academic purposes and prohibits commercial use that consistent with the licenses of our sampled datasets.
> > >
> > > As for the safety concerns related to text and images in our benchmark, we have conducted comprehensive safety checks as documented in Appendix C.2. Our analysis covers both image and text content. Only a very small portion of images were classified as NSFW, and we noted that the classification model occasionally produces false positives, as demonstrated in Figures 21 and 23. Additionally, we have addressed the trustworthiness concerns of interleaved generative models in Appendix A.
> > >
> > > ---
> > > Thank you again for taking the time to review our paper. Your feedback has helped us improve both the clarity and quality of our work.

---

> > > ### Comment · Reviewer_ASqC · 2024-11-20
> > > **Response to the authors**
> > >
> > > Thanks for the comprehensive response. I do acknowledge the popularity of using MLLM as a judge to evaluation. And thanks for providing the additional results in the setting that inferencing without visual inputs. I think this is a good paper and have raised my rating

---

> > > > ### Author Response · Authors · 2024-11-20
> > > > **Thanks!**
> > > >
> > > > We sincerely appreciate your time and dedication in reviewing our work, and are truly delighted by your strong endorsement of our research.

---

### Official Review · Reviewer_jj1D · 2024-11-09

**Soundness:** 2
**Presentation:** 3
**Contribution:** 3
**Rating:** 6
**Confidence:** 3

**Summary:**

This paper presents a multi-level evaluation framework, INTERLEAVED SCENE GRAPH (ISG), along with a benchmark dataset, ISG-BENCH, to address the challenges of evaluating multimodal interleaved text-and-image generation tasks. Additionally, the authors propose ISG-AGENT, a compositional generation framework designed to explore the upper limits of interleaved generation with a structured agent-based workflow.

**Strengths:**

1. Propose a novel ISG framework and ISG-BENCH benchmark dataset that fill a notable gap in the evaluation of multimodal interleaved generation.
2. The multi-level evaluation in ISG, combining visual question answering (VQA) with reasoning-based questions, offers a fine-grained approach to assess the structure and quality of generated content in complex multimodal tasks.
3. The creation of ISG-BENCH, covering 21 multimodal generation tasks, provides a standardized dataset for researchers.

**Weaknesses:**

1. The description of the methodology could be more detailed, for example, in terms of the experimental setup.
2. While ISG-AGENT performs well across tasks, it still exhibits notable shortcomings in specific areas. A more detailed discussion of these limitations and suggestions for future improvement would be beneficial in the conclusion.

**Questions:**

Please refer to weakness.

---

> ### Author Response · Authors · 2024-11-19
> **Rebuttal by Authors**
>
> Thank you very much for your valuable feedback. We apologize for any confusion caused by certain details in the paper. We will address each of your concerns and provide explanations to help you better understand the contributions of this paper step by step:
>
> ---
> **Q1:** The description of the methodology could be more detailed, for example, in terms of the experimental setup.
>
> **A1:** Thank you for your valuable suggestions. We have revised our method section by removing unnecessary notations and formulas, and added more examples to make our paper more accessible and easier to read. Regarding the experimental setup, we have moved some detailed settings from the Appendix to the experiment setup to make it clearer.
>
> ---
> **Q2:** Discussion of shortcoming of ISG-Agent.
>
> **A2:** Thank you for your suggestions. We acknowledge that as a compositional agent framework, ISG-Agent faces challenges in terms of more computing consumption compared to unified models and simple composition approaches. However, serving as a baseline, the consumption of ISG-Agent is acceptable. Additionally, we observe that ISG performs less effectively than simpler compositional frameworks (e.g., GPT-4o & SD3) in language-dominated VQA tasks. This limitation demonstrates that current SOTA tools do not always generate outputs that precisely align with query requirements, leading to suboptimal overall results. Therefore, improving tool selection and integration remains an important direction for future exploration in ISG-Agent.
>
> ---
> Thank you again for taking the time to review our paper. Your feedback has helped us improve both the clarity and quality of our work.

---

### Meta-Review · Area_Chair_SjrU · 2024-12-20

**Metareview:**

This paper presents a unified approach for evaluating interleaved multimodal text-and-image generation using a novel framework called INTERLEAVED SCENE GRAPH (ISG). Meanwhile, the paper presents a new (named ISG-BENCH),  which contains 1150 samples covering 21 generation scenarios. The benchmark has multiple-level of evaluation including structure, block, image, and holistic levels. It also introduces ISG-AGENT, a compositional agent framework designed to explore the limits of interleaved generation. Results show that existing models fall short of the benchmark and ISO-Agent performs significantly better than baseline models.

Strengths:
+ This paper presents the first benchmark to evaluate the models that can generate interleaved text-and-image models.
+ The design of ISG-Agent is reasonable and novel, achieving state-of-the-art performance on the ISG-Bench. The introduce of Planning, Tool-usage, and refinement works well in practice.
+ The paper is well-written and organized, with sufficient discussion on the evaluation pipeline, related works, and evaluation categories.

Weaknesses:
+ More details about the experiments (such as setup, hyperparameter settings), and some human ablation studies are expected.
+ More examples on the evaluation questions and category examples are expected.
+ More detailed discussion about the limitations are needed.

**Additional Comments On Reviewer Discussion:**

After the rebuttal, all reviewers provided positive ratings, and most of the concerns have been addressed.

---

### Decision · Program_Chairs · 2025-01-22

Accept (Spotlight)